# NAS-Bench-x11 and the Power of Learning Curves

**Shen Yan**[*1], **Colin White**[*2], **Yash Savani**[3], **Frank Hutter**[4,5]
[1] Michigan State University, [2] Abacus.AI, [3] Carnegie Mellon University,
[4] University of Freiburg, [5] Bosch Center for Artificial Intelligence

## Abstract

While early research in neural architecture search (NAS) required extreme computational resources, the recent releases of tabular and surrogate benchmarks have greatly increased the speed and reproducibility of NAS research. However, two of the most popular benchmarks do not provide the full training information for each architecture. As a result, on these benchmarks it is not possible to run many types of multi-fidelity techniques, such as learning curve extrapolation, that require evaluating architectures at arbitrary epochs. In this work, we present a method using singular value decomposition and noise modeling to create surrogate benchmarks, NAS-Bench-111, NAS-Bench-311, and NAS-Bench-NLP11, that output the full training information for each architecture, rather than just the final validation accuracy. We demonstrate the power of using the full training information by introducing a learning curve extrapolation framework to modify single-fidelity algorithms, showing that it leads to improvements over popular single-fidelity algorithms which claimed to be state-of-the-art upon release. Our code and pretrained models are available at `https://github.com/automl/nas-bench-x11`.

## 1 Introduction

In the past few years, algorithms for neural architecture search (NAS) have been used to automatically find architectures that achieve state-of-the-art performance on various datasets [82, 53, 38, 13]. In 2019, there were calls for reproducible and fair comparisons within NAS research [34, 57, 75, 36] due to both the lack of a consistent training pipeline between papers and experiments with not enough trials to reach statistically significant conclusions. These concerns spurred the release of tabular benchmarks, such as NAS-Bench-101 [76] and NAS-Bench-201 [11], created by fully training all architectures in search spaces of size $423\,624$ and $6\,466$, respectively. These benchmarks allow researchers to easily simulate NAS experiments, making it possible to run fair NAS comparisons and to run enough trials to reach statistical significance at very little computational cost or carbon emissions [17]. Recently, to extend the benefits of tabular NAS benchmarks to larger, more realistic NAS search spaces which cannot be evaluated exhaustively, it was proposed to construct *surrogate benchmarks* [60]. The first such surrogate benchmark is NAS-Bench-301 [60], which models the DARTS [38] search space of size $10^{18}$ architectures. It was created by fully training $60\,000$ architectures (both drawn randomly and chosen by top NAS methods) and then fitting a surrogate model that can estimate the performance of all of the remaining architectures. Since 2019, dozens of papers have used these NAS benchmarks to develop new algorithms [67, 55, 73, 64, 59].

An unintended side-effect of the release of these benchmarks is that it became significantly easier to devise *single fidelity* NAS algorithms: when the NAS algorithm chooses to evaluate an architecture, the architecture is fully trained and only the validation accuracy at the final epoch of training is outputted. This is because NAS-Bench-301 only contains the architectures' accuracy at epoch 100,

---

*Equal contribution. Correspondence to `yanshen6@msu.edu`, `colin@abacus.ai`, `ysavani@cs.cmu.edu`, `fh@cs.uni-freiburg.de`.

35th Conference on Neural Information Processing Systems (NeurIPS 2021).

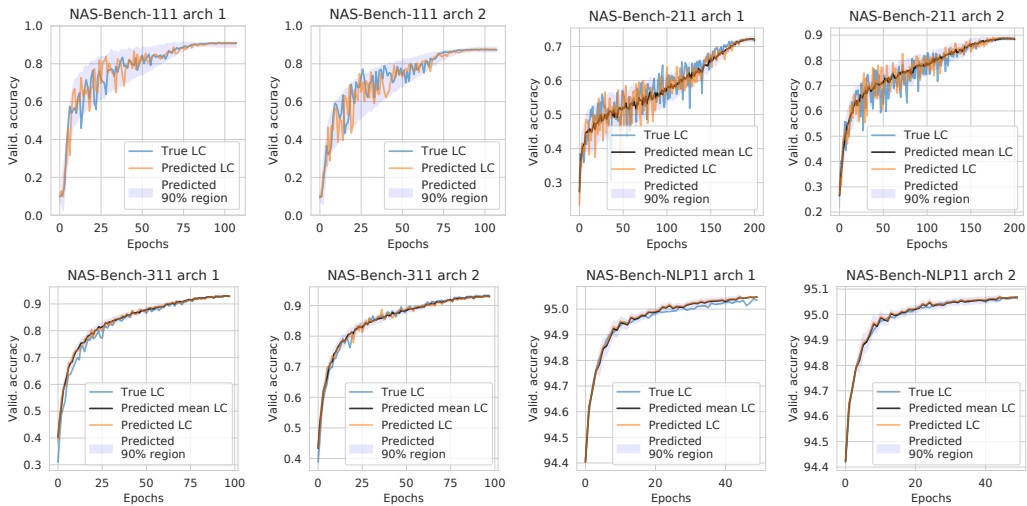

Figure 1: Each image shows the true learning curve vs. the learning curve predicted by our surrogate models, with and without predicted noise modeling. We also plot the 90% confidence interval of the predicted noise distribution. We plot two architectures each for NAS-Bench-111, NAS-Bench-211, NAS-Bench-311, and NAS-Bench-NLP11.

and NAS-Bench-101 only contains the accuracies at epochs 4, 12, 36, and 108 (allowing single fidelity or very limited multi-fidelity approaches). NAS-Bench-201 does allow queries on the entire learning curve (every epoch), but it is smaller in size ($6\,466$) than NAS-Bench-101 ($423\,624$) or NAS-Bench-301 ($10^{18}$). In a real world experiment, since training architectures to convergence is computationally intensive, researchers will often run *multi-fidelity* algorithms: the NAS algorithm can train architectures to any desired epoch. Here, the algorithm can make use of speedup techniques such as learning curve extrapolation (LCE) [63, 8, 1, 28] and successive halving [35, 14, 32, 29]. Although multi-fidelity techniques are often used in the hyperparameter optimization community [20, 21, 35, 14, 63], they have been under-utilized by the NAS community in the last few years.

In this work, we fill in this gap by releasing NAS-Bench-111, NAS-Bench-311, and NAS-Bench-NLP11, surrogate benchmarks with full learning curve information for train, validation, and test loss and accuracy for all architectures, significantly extending NAS-Bench-101, NAS-Bench-301, and NAS-Bench-NLP [30], respectively. With these benchmarks, researchers can easily incorporate multi-fidelity techniques, such as early stopping and LCE into their NAS algorithms. Our technique for creating these benchmarks can be summarized as follows. We use a training dataset of architectures (drawn randomly and chosen by top NAS methods) with good coverage over the search space, along with full learning curves, to fit a model that predicts the full learning curves of the remaining architectures. We employ three techniques to fit the model: *(1)* dimensionality reduction of the learning curves, *(2)* prediction of the top singular value coefficients, and *(3)* noise modeling. These techniques can be used in the future to create new NAS benchmarks as well. To ensure that our surrogate benchmarks are highly accurate, we report statistics such as Kendall Tau rank correlation and Kullback Leibler divergence between ground truth learning curves and predicted learning curves on separate test sets. See Figure 1 for examples of predicted learning curves on the test sets.

To demonstrate the power of using the full learning curve information, we present a framework for converting single-fidelity NAS algorithms into multi-fidelity algorithms using LCE. We apply our framework to popular single-fidelity NAS algorithms, such as regularized evolution [53], local search [68], and BANANAS [67], all of which claimed state-of-the-art upon release, showing that they can be further improved across four search spaces. Finally, we also benchmark multi-fidelity algorithms such as Hyperband [35] and BOHB [14] alongside single-fidelity algorithms. Overall, our work bridges the gap between different areas of AutoML and will allow researchers to easily develop effective multi-fidelity and LCE techniques in the future. To promote reproducibility, we release our code and we follow the NAS best practices checklist [36], providing the details in Appendix A.

**Our contributions.** We summarize our main contributions below.

Table 1: Overview of existing NAS benchmarks. We introduce NAS-Bench-111, -311, and -NLP11.

| Benchmark | Size | Queryable | Based on | Full train info |
|---|---|---|---|---|
| NAS-Bench-101 | 423k | ✓ | | ✗ |
| NAS-Bench-201 | 6k | ✓ | | ✓ |
| NAS-Bench-301 | $10^{18}$ | ✓ | DARTS | ✗ |
| NAS-Bench-NLP | $10^{53}$ | ✗ | | ✗ |
| NAS-Bench-ASR | 8k | ✓ | | ✓ |
| NAS-Bench-111 | 423k | ✓ | NAS-Bench-101 | ✓ |
| NAS-Bench-311 | $10^{18}$ | ✓ | DARTS | ✓ |
| NAS-Bench-NLP11 | $10^{53}$ | ✓ | NAS-Bench-NLP | ✓ |

- We develop a technique to create surrogate NAS benchmarks that include the full training information for each architecture, including train, validation, and test loss and accuracy learning curves. This technique can be used to create future NAS benchmarks on any search space.
- We apply our technique to create NAS-Bench-111, NAS-Bench-311, and NAS-Bench-NLP11, which allow researchers to easily develop multi-fidelity NAS algorithms that achieve higher performance than single-fidelity techniques.
- We present a framework for converting single-fidelity NAS algorithms into multi-fidelity NAS algorithms using learning curve extrapolation, and we show that our framework allows popular state-of-the-art NAS algorithms to achieve further improvements.

## 2   Related Work

NAS has been studied since at least the late 1980s [43, 26, 62] and has recently seen a resurgence [82, 44, 52, 22, 53, 18]. Weight sharing algorithms have become popular due to their computational efficiency [2, 38, 10, 79, 77, 51, 78]. Recent advances in performance prediction [65, 46, 59, 74, 39, 69, 55, 67] and other iterative techniques [14, 45] have reduced the runtime gap between iterative and weight sharing techniques. For detailed surveys on NAS, we suggest referring to [13, 71].

**Learning curve extrapolation.**   Several methods have been proposed to estimate the final validation accuracy of a neural network by extrapolating the learning curve of a partially trained neural network. Techniques include fitting the partial curve to an ensemble of various parametric functions [8], predicting the performance based on the derivatives of the learning curves of partially trained neural network configurations [1], summing the training losses [54], using the basis functions as the output layer of a Bayesian neural network [28], using previous learning curves as basis function extrapolators [4], using the positive-definite covariance kernel to capture a variety of training curves [63], or using a Bayesian recurrent neural network [15]. While in this work we focus on multi-fidelity optimization utilizing learning curve-based extrapolation, another main category of methods lie in bandit-based algorithm selection [35, 14, 29, 19, 40], and the fidelities can be further adjusted according to the previous observations or a learning rate scheduler [20, 21, 27].

**NAS benchmarks.**   NAS-Bench-101 [76], a tabular NAS benchmark, was created by defining a search space of size $423\,624$ unique architectures and then training all architectures from the search space on CIFAR-10 until 108 epochs. However, the train, validation, and test accuracies are only reported for epochs 4, 12, 36, and 108, and the training, validation, and test losses are not reported. NAS-Bench-1shot1 [80] defines a subset of the NAS-Bench-101 search space that allows one-shot algorithms to be run. NAS-Bench-201 [11] contains $15\,625$ architectures, of which $6\,466$ are unique up to isomorphisms. It comes with full learning curve information on three datasets: CIFAR-10 [31], CIFAR-100 [31], and ImageNet16-120 [7]. Recently, NAS-Bench-201 was extended to NATS-Bench [9] which searches over architecture size as well as architecture topology.

Virtually every published NAS method for image classification in the last three years evaluates on the DARTS search space with CIFAR-10 [61]. The DARTS search space [38] consists of $10^{18}$ neural architectures, making it computationally prohibitive to create a tabular benchmark. To overcome this fundamental limitation and query architectures in this much larger search space, NAS-Bench-301 [60]

evaluates various regression models trained on a sample of $60\,000$ architectures that is carefully created to cover the whole search space. The surrogate models allow users to query the validation accuracy (at epoch 100) and training time for any of the $10^{18}$ architectures in the DARTS search space. However, since the surrogates do not predict the entire learning curve, it is not possible to run multi-fidelity algorithms.

NAS-Bench-NLP [30] is a search space for language modeling tasks. The search space consists of $10^{53}$ architectures, of which $14\,322$ are evaluated on Penn Tree Bank [42], containing the training, validation, and test losses/accuracies from epochs 1 to 50. Since only $14\,322$ of $10^{53}$ architectures can be queried, this dataset cannot be directly used for NAS experiments. NAS-Bench-ASR [41] is a recent tabular NAS benchmark for speech recognition. The search space consists of $8\,242$ architectures with full learning curve information. For an overview of NAS benchmarks, see Table 1.

## 3 Creating Surrogate Benchmarks with Learning Curves

In this section, we describe our technique to create a surrogate model that outputs realistic learning curves, and then we apply this technique to create NAS-Bench-111, NAS-Bench-311, and NAS-Bench-NLP11. Our technique applies to any type of learning curve, including train/test losses and accuracies. For simplicity, the following presentation assumes validation accuracy learning curves.

### 3.1 General Technique

Given a search space $\mathcal{D}$, let $(\boldsymbol{x}_i, \boldsymbol{y}_i) \sim \mathcal{D}$ denote one datapoint, where $\boldsymbol{x}_i \in \mathbb{R}^d$ is the architecture encoding (e.g., one-hot adjacency matrix [76, 66]), and $\boldsymbol{y}_i \in [0, 1]^{E_{\max}}$ is a learning curve of validation accuracies drawn from a distribution $Y(\boldsymbol{x}_i)$ based on training the architecture for $E_{\max}$ epochs on a fixed training pipeline with a random initial seed. Without loss of generality, each learning curve $\boldsymbol{y}_i$ can be decomposed into two parts: one part that is deterministic and depends only on the architecture encoding, and another part that is based on the inherent noise in the architecture training pipeline. Formally, $\boldsymbol{y}_i = \mathbb{E}[Y(\boldsymbol{x}_i)] + \boldsymbol{\epsilon}_i$, where $\mathbb{E}[Y(\boldsymbol{x}_i)] \in [0, 1]^{E_{\max}}$ is fixed and depends only on $\boldsymbol{x}_i$, and $\boldsymbol{\epsilon}_i \in [0, 1]^{E_{\max}}$ comes from a noise distribution $Z_i$ with expectation 0 for all epochs. In practice, $\mathbb{E}[Y(\boldsymbol{x}_i)]$ can be estimated by averaging a large set of learning curves produced by training architecture $\boldsymbol{x}_i$ with different initial seeds. We represent such an estimate as $\bar{\boldsymbol{y}}_i$.

Our goal is to create a surrogate model that takes as input any architecture encoding $\boldsymbol{x}_i$ and outputs a distribution of learning curves that mimics the ground truth distribution. We assume that we are given two datasets, $\mathcal{D}_{\text{train}}$ and $\mathcal{D}_{\text{test}}$, of architecture and learning curve pairs. We use $\mathcal{D}_{\text{train}}$ (often size $> 10\,000$) to train the surrogate, and we use $\mathcal{D}_{\text{test}}$ for evaluation. We describe the process of creating $\mathcal{D}_{\text{train}}$ and $\mathcal{D}_{\text{test}}$ for specific search spaces in the next section. In order to predict a learning curve distribution for each architecture, we split up our approach into two separate processes: we train a model $f : \mathbb{R}^d \to [0, 1]^{E_{\max}}$ to predict the deterministic part of the learning curve, $\bar{\boldsymbol{y}}_i$, and we train a noise model $p_{\phi}(\boldsymbol{\epsilon} \mid \bar{\boldsymbol{y}}, \boldsymbol{x})$, parameterized by $\phi$, to simulate the random draws from $Z_i$.

**Surrogate model training.** Training a model $f$ to predict mean learning curves is a challenging task, since the training datapoints $\mathcal{D}_{\text{train}}$ consist only of a single (or few) noisy learning curve(s) $\boldsymbol{y}_i$ for each $\boldsymbol{x}_i$. Furthermore, $E_{\max}$ is typically length 100 or larger, meaning that $f$ must predict a high-dimensional output. We propose a technique to help with both of these challenges: we use the training data to learn compression and decompression functions $c_k : [0, 1]^{E_{\max}} \to [0, 1]^k$ and $d_k : [0, 1]^k \to [0, 1]^{E_{\max}}$, respectively, for $k \ll E_{\max}$. The surrogate is trained to predict *compressed* learning curves $c_k(\boldsymbol{y}_i)$ of size $k$ from the corresponding architecture encoding $\boldsymbol{x}_i$, and then each prediction can be reconstructed to a full learning curve using $d_k$. A good compression model should not only cause the surrogate prediction to become significantly faster and simpler, but should also reduce the noise in the learning curves, since it would only save the most important information in the compressed representations. That is, $(d_k \circ c_k)(\boldsymbol{y}_i)$ should be a less noisy version of $\boldsymbol{y}_i$. Therefore, models trained on $c_k(\boldsymbol{y}_i)$ tend to have better generalization ability and do not overfit to the noise in individual learning curves.

We test two compression techniques: singular value decomposition (SVD) [16] and variational autoencoders (VAEs) [25], and we show later that SVD performs better. We give the details of SVD here and describe the VAE compression algorithm in Appendix C. Formally, we take the singular value decomposition of a matrix $S$ of dimension $(|\mathcal{D}_{\text{train}}|, E_{\max})$ created by stacking together the

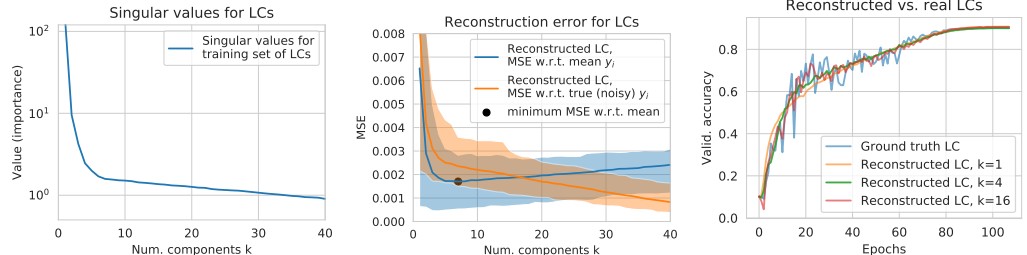

Figure 2: The singular values from the SVD decomposition of the learning curves (LC) (left). The MSE of a reconstructed LC, showing that $k = 6$ is closest to the true mean LC, while larger values of $k$ overfit to the noise of the LC (middle). An LC reconstructed using different values of $k$ (right).

learning curves from all architectures in $\mathcal{D}_{\text{train}}$. Performing the truncated SVD on the learning curve matrix $S$ allow us to create functions $c_k$ and $d_k$ that correspond to the optimal linear compression of $S$. In Figure 2 (left), we see that for architectures in the NAS-Bench-101 search space, there is a steep dropoff of importance after the first six singular values, which intuitively means that most of the information for each learning curve is contained in its first six singular values. In Figure 2 (middle), we compute the mean squared error (MSE) of the reconstructed learning curves $(d_k \circ c_k)(\boldsymbol{y}_i)$ compared to a test set of ground truth learning curves averaged over several initial seeds (approximating $\mathbb{E}[Y(\boldsymbol{x}_i)]$). The lowest MSE is achieved at $k = 6$, which implies that $k = 6$ is the value where the compression function minimizes reconstruction error without overfitting to the noise of individual learning curves. We further validate this in Figure 2 (right) by plotting $(d_k \circ c_k)(\boldsymbol{y}_i)$ for different values of $k$. Now that we have compression and decompression functions, we train a surrogate model with $\boldsymbol{x}_i$ as features and $c_k(\boldsymbol{y}_i)$ as the label, for architectures in $\mathcal{D}_{\text{train}}$. We test LGBoost [23], XGBoost [5], and MLPs for the surrogate model.

**Noise modeling.** The final step for creating a realistic surrogate benchmark is to add a noise model, so that the outputs are *noisy* learning curves. We first create a new dataset of predicted $\epsilon_i$ values, which we call residuals, by subtracting the reconstructed mean learning curves from the real learning curves in $\mathcal{D}_{\text{train}}$. That is, $\hat{\boldsymbol{\epsilon}}_i = \boldsymbol{y}_i - (d_k \circ c_k)(\boldsymbol{y}_i)$ is the residual for the $i$th learning curve. Since the training data only contains one (or few) learning curve(s) per architecture $\boldsymbol{x}_i$, it is not possible to accurately estimate the distribution $Z_i$ for each architecture without making further assumptions. We assume that the noise comes from an isotropic Gaussian distribution, and we consider two other noise assumptions: *(1)* the noise distribution is the same for all architectures (to create baselines, or for very homogeneous search spaces), and *(2)* for each architecture, the noise in a small window of epochs are iid. In Appendix C, we estimate the extent to which all of these assumptions hold true. Assumption *(1)* suggests two potential (baseline) noise models: *(i)* a simple sample standard deviation statistic, $\boldsymbol{\sigma} \in \mathbb{R}_+^{E_{\max}}$, where

$$\sigma_j = \sqrt{\frac{1}{|\mathcal{D}_{\text{train}}| - 1} \sum_{i=1}^{|\mathcal{D}_{\text{train}}|} \hat{\epsilon}_{i,j}^2}.$$

To sample the noise using this model, we sample from $\mathcal{N}(\boldsymbol{0}, \text{diag}(\boldsymbol{\sigma}))$. *(ii)* The second model is a Gaussian kernel density estimation (GKDE) model [58] trained on the residuals to create a multivariate Gaussian kernel density estimate for the noise. Assumption *(2)* suggests one potential noise model: *(iii)* a model that is trained to estimate the distribution of the noise over a window of epochs of a specific architecture. For each architecture and each epoch, the model is trained to estimate the sample standard deviation of the residuals within a window centered around that epoch.

Throughout this section, we described two compression methods (SVD and VAE), three surrogate models (LGB, XGB, and MLP), and three noise models (stddev, GKDE, and sliding window), for a total of eighteen disctinct approaches. In the next section, we describe the methods we will use to evaluate these approaches.

## 3.2 Surrogate Benchmark Evaluation

The performance of a surrogate benchmark depends on factors such as the size of the training set of architecture + learning curve pairs, the length of the learning curves, and the extent to which the distribution of learning curves satisfy the noise assumptions discussed in the previous section. It is important to thoroughly evaluate surrogate benchmarks before their use in NAS research, and so we present a number of different methods for evaluation, which we will use to evaluate the surrogate benchmarks we create in the next section.

We evaluate both the predicted mean learning curves and the predicted noise distributions using held-out test sets $\mathcal{D}_{\text{test}}$. To evaluate the mean learning curves, we compute the coefficient of determination ($R^2$) [70] and Kendall Tau (KT) rank correlation [24] between the set of predicted mean learning curves and the set of true learning curves, both at the final epoch and averaged over all epochs. To measure KT rank correlation for a specific epoch $n$, we find the number of concordant, $P$, and discordant, $Q$, pairs of predicted and true learning curve values for that epoch. The number of concordant pairs is given by the number of pairs, $((\hat{y}_{i,n}, y_{i,n}), (\hat{y}_{j,n}, y_{j,n}))$, where either both $\hat{y}_{i,n} > \hat{y}_{j,n}$ and $y_{i,n} > y_{j,n}$, or both $\hat{y}_{i,n} < \hat{y}_{j,n}$ and $y_{j,n} < y_{i,n}$. We can then calculate $KT = \frac{P-Q}{P+Q}$. While these metrics can be used to compare surrogate predictions and have been used in prior work [60], the rank correlation values are affected by the inherent noise in the learning curves - even an oracle would have $R^2$ and KT values smaller than 1.0 because architecture training is noisy. In order to give reference values, we also compute the KT of the set of true learning curves averaged over 2, 3, or 4 independent random seeds.

The next metric to be used in Section 3.3 is the Kullback Leibler (KL) divergence between the ground truth distribution of noisy learning curves, and the predicted distribution of noisy learning curves on a test set. Since we can only estimate the ground truth distribution, we assume the ground truth is an isotropic Gaussian distribution. Then we measure the KL divergence between the true and predicted learning curves for architecture $i$ by the following formula:

$$D_{KL}(\boldsymbol{y}_i || \hat{\boldsymbol{y}}_i) = \frac{1}{2E_{\max}} \left[ \log \frac{|\Sigma_{\hat{\boldsymbol{y}}_i}|}{|\Sigma_{\boldsymbol{y}_i}|} - E_{\max} + (\boldsymbol{\mu}_{\boldsymbol{y}_i} - \boldsymbol{\mu}_{\hat{\boldsymbol{y}}_i})^T \Sigma_{\hat{\boldsymbol{y}}_i}^{-1} (\boldsymbol{\mu}_{\boldsymbol{y}_i} - \boldsymbol{\mu}_{\hat{\boldsymbol{y}}_i}) + tr \left\{ \Sigma_{\hat{\boldsymbol{y}}_i}^{-1} \Sigma_{\boldsymbol{y}_i} \right\} \right]$$

where $\Sigma_{\{\boldsymbol{y}_i, \hat{\boldsymbol{y}}_i\}}$ is a diagonal matrix with the entries $\Sigma_{\{\boldsymbol{y}_i, \hat{\boldsymbol{y}}_i\}_{k,k}}$ representing the sample variance of the $k$-th epoch, and $\boldsymbol{\mu}_{\{\boldsymbol{y}_i, \hat{\boldsymbol{y}}_i\}}$ is the sample mean for either learning curves $\{\boldsymbol{y}_i, \hat{\boldsymbol{y}}_i\}$.

The final metric is the probability of certain anomalies which we call *spike anomalies*. Even if the KL divergences between the surrogate benchmark distributions and the ground-truth distributions are low, anomalies in the learning curves can still throw off NAS algorithms. For example, there may be anomalies that cause some learning curves to have a much higher maximum validation accuracy than their final validation accuracy. In order to test for these anomalies, first we compute the largest value $x$ such that there are fewer than 5% of learning curves whose maximum validation accuracy is $x$ higher than their final validation accuracy, on the real set of learning curves. Then, we compute the percentage of surrogate learning curves whose maximum validation accuracy is $x$ higher than their final validation accuracy. The goal is for this value to be close to 5%.

## 3.3 Creating NAS-Bench-111, -211, -311, and -NLP11

Now we describe the creation of NAS-Bench-111, NAS-Bench-311, and NAS-Bench-NLP11. We also create NAS-Bench-211 purely to evaluate our technique (since NAS-Bench-201 already has complete learning curves). Our code and pretrained models are available at https://github.com/automl/nas-bench-x11. As described above, we test two different compression methods (SVD, VAE), three different surrogate models (LGB, XGB, MLP), and three different noise models (stddev, GKDE, sliding window) for a total of eighteen distinct approaches. See Appendix C for a full ablation study and Table 2 for a summary using the best techniques for each search space. See Figure 1 for a visualization of predicted learning curves from the test set of each search space using these models.

First, we describe the creation of NAS-Bench-111. Since the NAS-Bench-101 tabular benchmark [76] consists only of accuracies at epochs 4, 12, 36, and 108 (and without losses), we train a new set of architectures and save the full learning curves. Similar to prior work [12, 60], we sample a set of architectures with good overall coverage while also focusing on the high-performing regions exploited

Table 2: Evaluation of the surrogate benchmarks on test sets. For NAS-Bench-111 and NAS-Bench-NLP11, we use architecture accuracies as additional features to improve performance.

| Benchmark | Avg. $R^2$ | Final $R^2$ | Avg. KT | Final KT | Avg. KL | Final KL |
|---|---|---|---|---|---|---|
| NAS-Bench-111 | 0.529 | 0.630 | 0.531 | 0.645 | 2.016 | 1.061 |
| NAS-Bench-111 (w. accs) | 0.630 | 0.853 | 0.611 | 0.794 | 1.710 | 0.926 |
| NAS-Bench-311 | 0.779 | 0.800 | 0.728 | 0.788 | 0.905 | 0.600 |
| NAS-Bench-NLP11 | 0.327 | 0.292 | 0.449 | 0.416 | - | - |
| NAS-Bench-NLP11 (w. accs) | 0.906 | 0.882 | 0.862 | 0.820 | - | - |

by NAS algorithms. Specifically, we sample 861 architectures generated uniformly at random, 149 architectures generated by 30 trials of BANANAS, local search, and regularized evolution, and all 91 architectures which contain fewer than five nodes, for a total of 1101 architectures. We kept our training pipeline as close as possible to the original pipeline. See Appendix C for the full training details. We find that SVD-LGB-window and SVD-LGB-GKDE achieve the best performance. Since the tabular benchmark already exists, we can substantially improve the accuracy of our surrogate by using the accuracies from the tabular benchmark (at epochs 4, 12, 36, 108) as additional features along with the architecture encoding. This substantially improves the performance of the surrogate, as shown in Table 2. The large difference between the average KT and last epoch KT values show that the learning curves are very noisy (which is also evidenced in Figure 1). On a separate test set that contains two seeds evaluated per architecture, we show that the KT values for NAS-Bench-111 are roughly equivalent to those achieved by a 1-seed tabular benchmark (Appendix Table 4).

Next, we create NAS-Bench-311 by using the training data from NAS-Bench-301, which consists of 40 000 random architectures along with 26 000 additional architectures generated by evolution [53], Bayesian optimization [3, 67, 47], and one-shot [38, 72, 10, 6] techniques in order to achieve good coverage over the search space. SVD-LGB-GKDE achieves the best performance. The GKDE model performing well is consistent with prior work that notes DARTS is very homogeneous [75]. NAS-Bench-311 achieves average and last epoch KT values of 0.728 and 0.788, respectively. This is comparable to the last epoch KT of 0.817 reported by NAS-Bench-301 [60], despite optimizing for the full learning curve rather than only the final epoch. Furthermore, our KL divergences in Table 2 surpasses the top value of 16.4 reported by NAS-Bench-301 (for KL divergence, lower is better). In Appendix Table 5, we show that the mean model in NAS-Bench-311 achieves higher rank correlation even than a set of learning curves averaged over four random seeds, by using a separate test set from the NAS-Bench-301 dataset which evaluates 500 architectures with 5 seeds each. We also show that the percentage of spike anomalies for real vs. surrogate data (defined in the previous section) is 5% and 7.02%, respectively.

Finally, we create NAS-Bench-NLP11 by using the NAS-Bench-NLP dataset consisting of 14 322 architectures drawn uniformly at random. Due to the extreme size of the search space ($10^{53}$), we achieve an average and final epoch KT of 0.449 and 0.416, respectively. Since there are no architectures trained multiple times on the NAS-Bench-NLP dataset [30], we cannot compute the KL divergence or additional metrics. To create a stronger surrogate, we add the first three epochs of the learning curve as features in the surrogate. This improves the average and final epoch KT values to 0.862 and 0.820, respectively. To use this surrogate, any architecture to be predicted with the surrogate must be trained for three epochs. Note that the small difference between the average and last epoch KT values indicates that the learning curves have very little noise, which can also be seen in Figure 1. This technique can be used to improve the performance of other surrogates too, such as NAS-Bench-311. For NAS-Bench-311, we find that adding the validation accuracies from the first three epochs as additional epochs, improves the average and final epoch KT from 0.728 and 0.788 to 0.749 and 0.795, respectively.

We also create NAS-Bench-211 to further evaluate our surrogate creation technique (only for evaluation purposes, since NAS-Bench-201 already has complete learning curves). We train the surrogate on 90% of architectures from NAS-Bench-201 (14 062 architectures) and test on the remaining 10%. SVD-LGB-window achieves the best performance. The average and final KT values are 0.701 and 0.842, respectively, which is on par with a 1-seed tabular benchmark (see Table 6). We also show that the percentage of spike anomalies for real vs. surrogate data (defined in the previous section) is 5% and 10.14%, respectively.

## 4 The Power of Learning Curve Extrapolation

Now we describe a simple framework for converting single-fidelity NAS algorithms to multi-fidelity NAS algorithms using learning curve extrapolation techniques. We show that this framework is able to substantially improve the performance of popular algorithms such as regularized evolution [53], BANANAS [67], and local search [68, 48].

A single-fidelity algorithm is an algorithm which iteratively chooses an architecture based on its history, which is then fully trained to $E_{\max}$ epochs. To exploit parallel resources, many single-fidelity algorithms iteratively output several architectures at a time, instead of just one. In Algorithm 1, we present pseudocode for a generic single-fidelity algorithm. For example, for local search, `gen_candidates` would output the neighbors of the architecture with the best accuracy in `history`.

Our framework makes use of learning curve extrapolation (LCE) techniques [8, 1] to predict the final validation accuracies of all architecture choices after only training for a small number of epochs. See Algorithm 2. After each iteration of `gen_candidates`, only the architectures predicted by `LCE()` to be in the top percentage of validation accuracies of `history` are fully trained. For example, when the framework is applied to local search, in each iteration, all neighbors are trained to $E_{\text{few}}$ epochs, and only the most promising architectures are trained up to $E_{\max}$ epochs. This simple modification can substantially improve the runtime efficiency of popular NAS algorithms by weeding out unpromising architectures before they are fully trained.

Any LCE technique can be used, and in our experiments in Section 5, we use weighted probabilistic modeling (WPM) [8] and learning curve support vector regressor (LcSVR) [1]. The first technique, WPM [8], is a function that takes a partial learning curve as input, and then extrapolates it by fitting the learning curve to a set of parametric functions, using MCMC to sample the most promising fit. The second technique, LcSVR [1], is a model-based learning curve extrapolation technique: after generating an initial set of training architectures, a support vector regressor is trained to predict the final validation accuracy from the architecture encoding and partial learning curve.

---

**Algorithm 1** Single-Fidelity Algorithm

```
1: initialize history
2: while t < t_max :
3:    arches = gen_candidates(history)
4:    accs = train(arches, epoch=E_max)
5:    history.update(arches, accs)
6: Return arch with the highest acc
```

**Algorithm 2** LCE Framework

```
1: initialize history
2: while t < t_max :
3:    arches = gen_candidates(history)
4:    accs = train(arches, epoch=E_few)
5:    sorted_by_pred = LCE(arches, accs)
6:    arches = sorted_by_pred[:top_n]
7:    accs = train(arches, epoch=E_max)
8:    history.update(arches, accs)
9: Return arch with the highest acc
```

---

## 5 Experiments

In this section, we benchmark single-fidelity and multi-fidelity NAS algorithms, including popular existing single-fidelity and multi-fidelity algorithms, as well as algorithms created using our framework defined in the previous section. In the experiments, we use our three surrogate benchmarks defined in Section 3, as well as NAS-Bench-201.

**NAS algorithms.** For single-fidelity algorithms, we implemented random search (RS) [34], local search (LS) [68, 48], regularized evolution (REA) [53], and BANANAS [67]. For multi-fidelity bandit-based algorithms, we implemented Hyperband (HB) [35] and Bayesian optimization Hyperband (BOHB) [14]. For all methods, we use the original implementation whenever possible. See Appendix D for a description, implementation details, and hyperparameter details for each method. Finally, we use our framework from Section 4 to create six new multi-fidelity algorithms: BANANAS, LS, and REA are each augmented using WPM and LcSVR. This gives us a total of 12 algorithms.

**Experimental setup.** For each search space, we run each algorithm for a total wall-clock time that is equivalent to running 500 iterations of the single-fidelity algorithms for NAS-Bench-111 and

NAS-Bench-311, and 100 iterations for NAS-Bench-201 and NAS-Bench-NLP11. For example, the average time to train a NAS-Bench-111 architecture to 108 epochs is roughly $10^3$ seconds, so we set the maximum runtime on NAS-Bench-111 to roughly $5 \cdot 10^5$ seconds. We run 30 trials of each NAS algorithm and compute the mean and standard deviation.

**Results.** We evaluate BANANAS, LS, and REA compared to their augmented WPM and SVR versions in Figure 3 (NAS-Bench-311) and Appendix D (all other search spaces). Across four search spaces, we see that WPM and SVR improve all algorithms in almost all settings. The improvements are particularly strong for the larger NAS-Bench-111 and NAS-Bench-311 search spaces. We also see that for each single-fidelity algorithm, the LcSVR variant often outperforms the WPM variant. This suggests that model-based techniques for extrapolating learning curves are more reliable than extrapolating each learning curve individually, which has also been noted in prior work [69].

In Figure 4, we compare single- and multi-fidelity algorithms on four search spaces, along with the three SVR-based algorithms from Figure 3. Across all search spaces, an SVR-based algorithm is the top-performing algorithm. Specifically, BANANAS-SVR performs the best on NAS-Bench-111, NAS-Bench-311, and NAS-Bench-NLP11, and LS-SVR performs the best on NAS-Bench-201. Note that HB and BOHB may not perform well on search spaces with low correlation between the relative rankings of architectures using low fidelities and high fidelities (such as NAS-Bench-101 [76] and NAS-Bench-201 [11]) since HB-based methods will predict the final accuracy of partially trained architectures directly from the last trained accuracy (i.e., extrapolating the learning curve as a constant after the last seen accuracy). On the other hand, SVR-based approaches use a model that can learn more complex relationships between accuracy at an early epoch vs. accuarcy at the final epoch, and are therefore more robust to this type of search space.

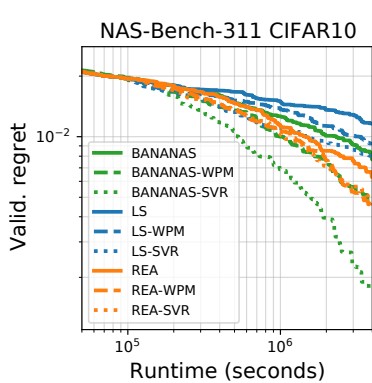

Figure 3: LCE Framework applied to single-fidelity algorithms on NAS-Bench-311.

In Appendix D, we perform an ablation study on the epoch at which the SVR and WPM methods start extrapolating in our framework (i.e., we ablate $E_{\text{few}}$ from Algorithm 2). We find that for most search spaces, running SVR and WPM based NAS methods by starting the LCE at roughly $20\%$ of the total number of epochs performs the best. Any earlier, and there is not enough information to accurately extrapolate the learning curve. Any later, and the LCE sees diminishing returns because less time is saved by early stopping the training.

## 6 Societal Impact

Our hope is that our work will make it quicker and easier for researchers to run fair experiments and give reproducible conclusions. In particular, the surrogate benchmarks allow AutoML researchers to develop NAS algorithms directly on a CPU, as opposed to using a GPU, which may decrease the carbon emissions from GPU-based NAS research [50, 17]. In terms of our proposed NAS speedups, these techniques are a level of abstraction away from real applications, but they can indirectly impact the broader society. For example, this work may facilitate the creation of new high-performing NAS techniques, which can then be used to improve various deep learning methods, both beneficial (e.g. algorithms that reduce $CO_2$ emissions), or harmful (e.g. algorithms that produce deep fakes).

## 7 Conclusions, Limitations, and Guidelines

In this work, we released three benchmarks for neural architecture search based on three popular search spaces, which substantially improve the capability of existing benchmarks due to the availability of the full learning curve for train/validation/test loss and accuracy for each architecture. Our techniques to generate these benchmarks, which includes singular value decomposition of the learning curve and noise modeling, can be used to model the full learning curve for future surrogate NAS benchmarks as well.

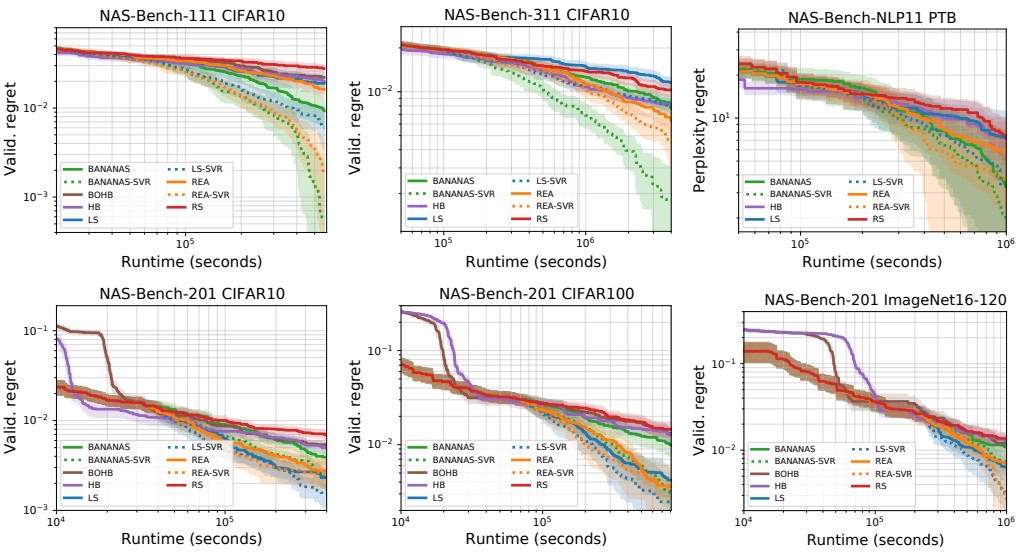

Figure 4: NAS results on six different combinations of search spaces and datasets. For every setting, an SVR augmented method performs best.

Furthermore, we demonstrated the power of the full learning curve information by introducing a framework that converts single-fidelity NAS algorithms into multi-fidelity NAS algorithms that make use of learning curve extrapolation techniques. This framework improves the performance of recent popular single-fidelity algorithms which claimed to be state-of-the-art upon release.

While we believe our surrogate benchmarks will help advance scientific research in NAS, a few guidelines and limitations are important to keep in mind. As with prior surrogate benchmarks [60], we give the following two caveats. *(1)* We discourage evaluating NAS methods that use the same internal techniques as those used in the surrogate model. For example, any NAS method that makes use of variational autoencoders or XGBoost should not be benchmarked using our VAE-XGB surrogate benchmark. *(2)* As the surrogate benchmarks are likely to evolve as new training data is added, or as better techniques for training a surrogate are devised, we recommend reporting the surrogate benchmark version number whenever running experiments. *(3)* When creating new surrogate benchmarks, before use in NAS, it is important to give a thorough evaluation such as the evaluation methods described in Section 3.2.

We also note the following strengths and limitations for specific benchmarks. Our NAS-Bench-111 surrogate benchmark gives strong mean performance even with just $1\,101$ architectures used as training data, due to the existence of the four extra validation accuracies from NAS-Bench-101 that can be used as additional features. We hope that all future tabular benchmarks will save the full learning curve information from the start so that the creation of an after-the-fact extended benchmark is not necessary.

Although all of our surrogates had strong mean performance, the noise model was strongest only for NAS-Bench-311, which had a training dataset of size 60k, as evidenced especially because of the rate of spike anomalies, described in Section 3.2. This can be mitigated by adding more training data.

Since the NAS-Bench-NLP11 surrogate benchmark achieves significantly stronger performance when the accuracy of the first three epochs are added as features, we recommend using this benchmark by training architectures for three epochs before querying the surrogate. Therefore, benchmarking NAS algorithms are slower than for NAS-Bench-111 and NAS-Bench-311, but NAS-Bench-NLP11 still offers a $15\times$ speedup compared to a NAS experiment without this benchmark. We still release the version of NAS-Bench-NLP11 that does not use the first three accuracies as features but with a warning that the observed NAS trends may differ from the true NAS trends. We expect that the performance of these benchmarks will improve over time, as data for more trained architectures become available.

## Acknowledgments and Disclosure of Funding

Work done while the first three authors were working at Abacus.AI. FH acknowledges support by the European Research Council (ERC) under the European Union Horizon 2020 research and innovation programme through grant no. 716721, BMBF grant DeToL, and TAILOR, a project funded by EU Horizon 2020 research and innovation programme under GA No 952215. We thank Kaicheng Yu for his discussions with this project.

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
