# A  NAS Best Practices Checklist

In the past few years, the NAS community has called for improving the reproducibility and fairness in experimental comparisons [34, 76, 75]. Recently, a NAS best practices checklist was released [36]. We answer each question from this checklist below.

1. **Best Practices for Releasing Code**

   For all experiments you report:

   (a) Did you release code for the training pipeline used to evaluate the final architectures? [N/A] We used the training pipelines from NAS-Bench-101, NAS-Bench- 301, and NAS-Bench-NLP, and the code for all three are already publicly available.

   (b) Did you release code for the search space [N/A] We used the search spaces from NAS-Bench-101, NAS-Bench- 301, and NAS-Bench-NLP, and the code for all three are already publicly available.

   (c) Did you release the hyperparameters used for the final evaluation pipeline, as well as random seeds? [N/A] As with prior work that use NAS-Bench-101, NAS-Bench- 301, and NAS-Bench-NLP, our final evaluation pipeline is identical to the training pipeline. Since we averaged over 30 trials of each experiment, we did not report random seeds.

   (d) Did you release code for your NAS method? [Yes] Our code is available at `https://github.com/automl/nas-bench-x11`.

   (e) Did you release hyperparameters for your NAS method, as well as random seeds? [Yes] Our code includes runner files with the same hyperparameters and seeds from our paper.

2. **Best practices for comparing NAS methods**

   (a) For all NAS methods you compare, did you use exactly the same NAS benchmark, including the same dataset (with the same training-test split), search space and code for training the architectures and hyperparameters for that code? [Yes] This is true automatically because we only used NAS benchmarks, which fix the training and evaluation protocols.

   (b) Did you control for confounding factors (different hardware, versions of DL libraries, different runtimes for the different methods)? [Yes] This is true automatically because we only used NAS Benchmarks, which fix the training and evaluation protocols.

   (c) Did you run ablation studies? [Yes] In Sections 4 and D.2, we run ablation studies for the our LCE framework. In Section C, we run ablation studies for our surrogate benchmarks.

   (d) Did you use the same evaluation protocol for the methods being compared? [Yes] This is true automatically because we only used NAS Benchmarks, which fix the training and evaluation protocols.

   (e) Did you compare performance over time? [Yes] We did compare performance over time.

   (f) Did you compare to random search? [Yes] We did compare to random search.

   (g) Did you perform multiple runs of your experiments and report seeds? [Yes] We ran 30 trials for each experiment.

   (h) Did you use tabular or surrogate benchmarks for in-depth evaluations? [Yes] We did use NAS benchmarks.

3. **Best practices for reporting important details**

   (a) Did you report how you tuned hyperparameters, and what time and resources this required? [Yes] We did report information on tuning hyperparameters, including an ablation study in Section D.2.

   (b) Did you report the time for the entire end-to-end NAS method (rather than, e.g., only for the search phase)? [Yes] Our results include the end-to-end NAS time.

   (c) Did you report all the details of your experimental setup? [Yes] We include all details of our experimental setup.

# B Related Work Continued

In this section, we give a more detailed discussion of related work (a superset of the related work discussed in Section 2).

NAS has been studied since at least the late 1980s [43, 26, 62] and has recently seen a resurgence [82, 44, 52, 22, 53, 18]. Early techniques included reinforcement learning [82, 52], regularized evolution [53], and Bayesian optimization [22]. Recently, weight sharing [52, 38] has become a popular approach to substantially speed up the runtime of NAS. In this approach, an over-parameterized supernetwork is trained, which can represent all architectures in the search space. Then all architectures in the search space can be evaluated using the shared weights. Some work has claimed that the shared weights are sometimes not effective at ranking architectures [57, 80, 81], however, weight sharing techniques still achieve strong overall NAS performance [79, 33].

Recently, many works have been devoted to performance prediction [65, 46, 59, 74, 39, 69, 55, 67] and multi-fidelity techniques [14] which has reduced the runtime gap between iterative and weight sharing techniques. For detailed surveys on NAS, see [13, 71]. The most widely used type of search space in prior work is the cell-based search space [83, 37], where the architecture search is over a relatively small directed acyclic graph representing an architecture.

**Learning curve extrapolation.** Several methods have been proposed to estimate the final validation accuracy of a neural network by extrapolating the learning curve of a partially trained neural network. Techniques include, fitting the partial curve to an ensemble of parametric functions [8], predicting the performance based on the partial trained neural network configurations [1], summing the training losses [54], using the basis functions as the output layer of a Bayesian neural network [28], using previous learning curves as basis function extrapolators [4], using the positive-definite covariance kernel to capture a variety of training curves [63], or using a Bayesian recurrent neural network [15]. While in this work we focus on multi-fidelity optimization utilizing learning curve-based extrapolation, another main category of methods lie in bandit-based algorithm selection [35, 14, 29, 19, 40], and the fidelities can be further adjusted according to the previous observations or a learning rate scheduler [20, 21, 27].

**NAS benchmarks.** NAS-Bench-101 [76], a tabular NAS benchmark, was created by defining a search space of size $423\,624$ unique architectures and then training all architectures from the search space on CIFAR-10 until 108 epochs. However, the train, validation, and test accuracies are only reported for epochs 4, 12, 36, and 108, and the train/valid/test losses are not reported. NAS-Bench-1shot1 [80] defines a subset of the NAS-Bench-101 search space that allows one-shot algorithms to be run. NAS-Bench-201 [11] contains $15\,625$ architectures, of which $6\,466$ are unique up to isomorphisms. It comes with full learning curve information on three datasets: CIFAR-10 [31], CIFAR-100 [31], and ImageNet16-120 [7]. Recently, NAS-Bench-201 was extended to NATS-Bench [9] which searches over architecture size as well as architecture topology.

Virtually every published NAS method for image classification in the last 3 years evaluates on the DARTS search space with CIFAR-10 [61]. The DARTS search space [38] consists of $10^{18}$ neural architectures, making it computationally prohibitive to create a tabular benchmark. To overcome this fundamental limitation and query architectures in this much larger search space, NAS-Bench-301 [60] evaluates various regression models trained on a sample of $60\,000$ architectures that is carefully created to cover the whole search space. The surrogate models allow users to query the validation accuracy (at epoch 100) and training time for any of the $10^{18}$ architectures in the DARTS search space. However, since the surrogates do not predict the entire learning curve, it is not possible to run multi-fidelity algorithms.

NAS-Bench-NLP [30] is a search space for language modeling tasks. The search space consists of $10^{53}$ LSTM-like architectures, of which $14\,322$ are evaluated on Penn Tree Bank [42], containing the training, validation, and test losses/accuracies from epochs 1 to 50. Since only $14\,322$ of $10^{53}$ architectures can be queried, this dataset cannot be directly used for NAS experiments. NAS-Bench-ASR [41] is a recent tabular NAS benchmark for speech recognition. The search space consists of $8\,242$ architectures with full learning curve information. For an overview of NAS benchmarks, see Table 1. We give a more detailed discussion of all related work in Appendix B.

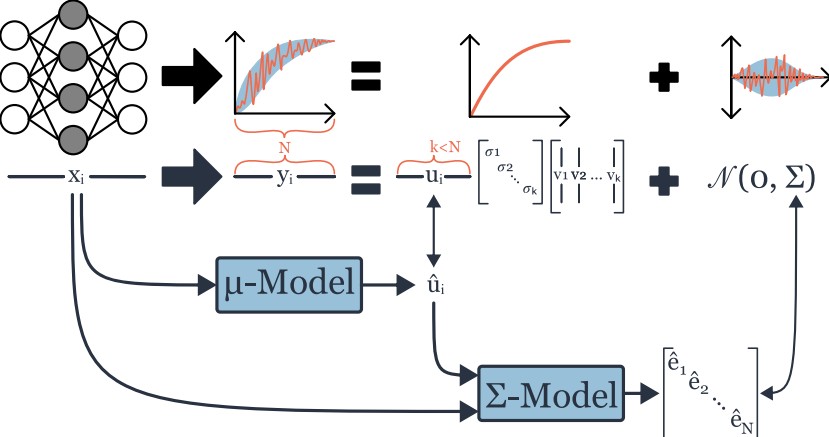

Figure 5: A summary of our approach to create surrogate benchmarks that output realistic learning curves. Compression and decompression functions are learned using the training set of learning curves (in the figure, SVD is shown, but a VAE can also be used). The compression also helps to de-noise the learning curves. A model ($\mu$-model) is trained to predict the compressed (de-noised) learning curves given the architecture encoding. A separate model ($\Sigma$-model) is trained to predict each learning curve's noise distribution, given the architecture encoding and predicted compressed learning curve. A realistic learning curve can then be outputted by decompressing the predicted learning curve and sampling noise from the noise distribution.

## C   Details from Section 3

In this section, we give more details from Section 3, and we present a full ablation study.

Recall the following notation, repeated from Section 3. Given a search space $\mathcal{D}$, let $(\boldsymbol{x}_i, \boldsymbol{y}_i) \sim \mathcal{D}$ denote one datapoint, where $\boldsymbol{x}_i \in \mathbb{R}^d$ is the architecture encoding, and $\boldsymbol{y}_i \in [0,1]^{E_{\max}}$ is a learning curve of validation accuracies drawn from a distribution $Y(\boldsymbol{x}_i)$ based on training the architecture for $E_{\max}$ epochs on a fixed training pipeline with a random initial seed. For each learning curve $\boldsymbol{y}_i$, we have $\boldsymbol{y}_i = \mathbb{E}[Y(\boldsymbol{x}_i)] + \boldsymbol{\epsilon}_i$, where $\mathbb{E}[Y(\boldsymbol{x}_i)] \in [0,1]^{E_{\max}}$ is fixed and depends only on $\boldsymbol{x}_i$, and $\boldsymbol{\epsilon}_i \in [0,1]^{E_{\max}}$ comes from a noise distribution $Z_i$ with expectation 0 for all epochs. In practice, $\mathbb{E}[Y(\boldsymbol{x}_i)]$ can be estimated by averaging a large set of learning curves produced by training architecture $\boldsymbol{x}_i$ with different initial seeds. We represent such an estimate as $\bar{\boldsymbol{y}}_i$. As explained in Section 3, we split the surrogate model creation into two parts: we train a model $f : \mathbb{R}^d \to [0,1]^{E_{\max}}$ to predict the deterministic part of the learning curve, $\bar{\boldsymbol{y}}_i$, and we train a noise model $p_\phi(\boldsymbol{\epsilon} \mid \bar{\boldsymbol{y}}, \boldsymbol{x})$, parameterized by $\phi$, to simulate the random draws from $Z_i$. See Figure 5 for a summary of our entire surrogate creation method (assuming SVD).

In Section 3, we described singular value decomposition (SVD) as a technique to create the compression and decompression functions $c_k : [0,1]^{E_{\max}} \to [0,1]^k$ and $d_k : [0,1]^k \to [0,1]^{E_{\max}}$, respectively, for $k \ll E_{\max}$, which aid in the creation of a model $f$. Now we describe our second technique for compression and decompression: a variational autoencoder (VAE) [25]. The VAE has the benefit over SVD that it is a non-linear dimensionality reduction technique. However, it is harder to train as it has more hyperparameters, as opposed to SVD which only has $k$ as a parameter. We fit a VAE model to the learning curves. We used a simple PyTorch [49] implementation of VAE which has four fully connected layers of 512 nodes, separated by ReLU, in both the encoder and decoder architecture. Finally, we add dropout of 0.2 and the Adam optimizer. We constrain the dimension of the bottleneck latent space the same amount as with SVD: $k = 5$. This creates a non-linear dimensionality reduction model. In Table 3, we see that the VAE does not perform as well as SVD.

As mentioned in Section 3, we try three different models for the main $\mu$-model (which predicts the compressed learning curves from the architecture encodings): LGBoost [23], XGBoost [5], and a standard multilayer perceptron (MLP). For LGBoost and XGBoost, we use the default parameters reported from NAS-Bench-301 [60]. For MLP, we use one layer with 64 nodes, with SGD with learning rate 0.001. We also tried five layers, which performed worse.

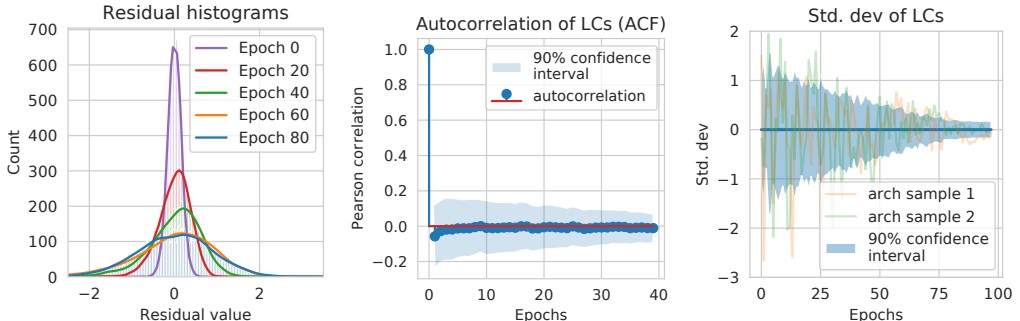

Figure 6: A plot of the residuals across all architectures for five different epochs (left). We see that the distributions are roughly Gaussian. A plot of the autocorrelation function (ACF) averaged over all training learning curves (middle). We see that there is only a small amount of autocorrelation. A plot of the 90% confidence intervals of the residuals at each epoch (right). All plots use the NAS-Bench-301 learning curve training set.

Finally, we consider three different noise models as described in Section 3. Recall that we create a new dataset of predicted $\epsilon_i$ values, which we call residuals, by subtracting the reconstructed mean learning curves from the real learning curves in $\mathcal{D}_{\text{train}}$. That is, $\hat{\epsilon}_i = \boldsymbol{y}_i - (d_k \circ c_k)(\boldsymbol{y}_i)$ is the residual for the $i$th learning curve. Our three noise models are based on two different assumptions: *(1)* the noise distribution is the same for all architectures, and *(2)* for each architecture, the noise in a small window of epochs are iid. Now we evaluate these assumptions. In Figure 6 (left), we plot the residuals from the NAS-Bench-301 training set at five different epochs, showing that the distributions are roughly Gaussian, across all architectures. Recall that noise model *(i)* is a simple standard deviation statistic computed for each epoch independently, across all architectures. In Figure 6 (middle), we plot the autocorrelation function (ACF) averaged over all training learning curves on NAS-Bench-301. We see that there is very little autocorrelation in the learning curves, which justifies the use of the first noise model. Recall that our second noise model uses Gaussian kernel density estimation (GKDE) [58] across all learning curves. This is essentially the same as the first noise model but with the ability to capture the small amount of autocorrelation present. Finally, recall that our our third noise model does not assume that the residual distribution is similar across all architectures. Instead, it estimates the standard deviation for each epoch using a sliding window of size 10 across the epochs for each architecture. See Figure 6 (right) for the 90% confidence intervals of the residuals at each epoch on NAS-Bench-301. Although the sliding window noise model has the benefit of capturing different distributions for different architectures, we see that the standard deviation steadily decreases as the epoch number increases, meaning that the noise in a small window of epochs are not perfectly iid.

In Table 3, we run a full ablation study by testing all eighteen combinations of {SVD, VAE}, {LGB, XGB, MLP}, and {GKDE, STD, window} for NAS-Bench-111 and NAS-Bench-311. Recall that, as explained in Section 3, the first four metrics evaluate only the prediction of the mean learning curves using a held-out test set, so the noise model has no effect on the first four metrics (average $R^2$, final $R^2$, average KT, and final KT). For the final two metrics (average KL and final KL), we use a test set of learning curves consisting of five seeds of architectures, so that we can estimate the KL divergence between the real learning curve distribution and the predicted distribution. Note that none of the NAS-Bench-NLP architectures were trained more than once, so we are unable to test the noise models for NAS-Bench-NLP11. Across NAS-Bench-111, NAS-Bench-311, and NAS-Bench-NLP11, we see that SVD-LGB performs substantially better than all of the other options for the model.

**Surrogate training details.** Finally, we give more details for the surrogate training. For NAS-Bench-111, as discussed in Section 3, we created a new set of trained architectures with the full learning curve information. We kept the training pipeline nearly the same as in the orginal NAS-Bench-101 repository. However, instead of the TPU v2 acceleator as in the original work, we used an RTX 3070. We needed to change the batch size from 256 to 200 to account for this hardware change, which we found had a negligible affect on the final accuracy. We trained 1101 new architectures and used this new set as the new "ground-truth" when training and evaluating NAS-Bench-111. As

Table 3: Evaluation of the surrogate benchmarks on test sets, with all combinations of models. For NAS-Bench-111 and NAS-Bench-NLP11, we use architecture accuracies as additional features to improve performance. As explained in Section C, no architectures in the NAS-Bench-NLP dataset were trained more than once, so we do not compute KL divergence for NAS-Bench-NLP11.

| Benchmark | Avg. $R^2$ | Final $R^2$ | Avg. KT | Final KT | Avg. KL | Final KL |
|---|---|---|---|---|---|---|
| NAS-Bench-111 | | | | | | |
| SVD-LGB-GKDE | **0.630** | **0.853** | **0.611** | **0.794** | **1.641** | 0.516 |
| SVD-LGB-STD | **0.630** | **0.853** | **0.611** | **0.794** | 2.768 | **0.383** |
| SVD-LGB-window | **0.630** | **0.853** | **0.611** | **0.794** | 24.402 | 3.303 |
| SVD-XGB-GKDE | 0.329 | 0.378 | 0.408 | 0.429 | 2.743 | 0.580 |
| SVD-XGB-STD | 0.329 | 0.378 | 0.408 | 0.429 | 4.867 | 0.503 |
| SVD-XGB-window | 0.329 | 0.378 | 0.408 | 0.429 | 38.457 | 16.172 |
| SVD-MLP-GKDE | 0.195 | 0.065 | 0.330 | 0.290 | 4.599 | 0.762 |
| SVD-MLP-STD | 0.195 | 0.065 | 0.330 | 0.290 | 8.417 | 0.848 |
| SVD-MLP-window | 0.195 | 0.065 | 0.330 | 0.290 | 82.180 | 15.711 |
| VAE-LGB-GKDE | 0.267 | 0.218 | 0.462 | 0.617 | 3.788 | 0.829 |
| VAE-LGB-STD | 0.267 | 0.218 | 0.462 | 0.617 | 6.866 | 0.972 |
| VAE-LGB-window | 0.267 | 0.218 | 0.462 | 0.617 | 53.866 | 19.820 |
| VAE-XGB-GKDE | 0.311 | 0.272 | 0.453 | 0.559 | 3.828 | 0.828 |
| VAE-XGB-STD | 0.311 | 0.272 | 0.453 | 0.559 | 6.940 | 0.969 |
| VAE-XGB-window | 0.311 | 0.272 | 0.453 | 0.559 | 55.654 | 19.614 |
| VAE-MLP-GKDE | 0.218 | 0.007 | 0.386 | 0.369 | 4.583 | 0.844 |
| VAE-MLP-STD | 0.218 | 0.007 | 0.386 | 0.369 | 8.386 | 1.001 |
| VAE-MLP-window | 0.218 | 0.007 | 0.386 | 0.369 | 83.481 | 19.091 |
| NAS-Bench-311 | | | | | | |
| SVD-LGB-GKDE | **0.779** | **0.800** | **0.728** | **0.788** | **0.503** | **0.548** |
| SVD-LGB-STD | **0.779** | **0.800** | **0.728** | **0.788** | 0.919 | 1.036 |
| SVD-LGB-window | **0.779** | **0.800** | **0.728** | **0.788** | 1.566 | 4.083 |
| SVD-XGB-GKDE | 0.522 | 0.546 | 0.607 | 0.654 | 1.783 | 3.272 |
| SVD-XGB-STD | 0.522 | 0.546 | 0.607 | 0.654 | 3.271 | 5.958 |
| SVD-XGB-window | 0.522 | 0.546 | 0.607 | 0.654 | 5.282 | 19.432 |
| SVD-MLP-GKDE | 0.564 | 0.549 | 0.573 | 0.603 | 15.727 | 29.057 |
| SVD-MLP-STD | 0.564 | 0.549 | 0.573 | 0.603 | 28.833 | 52.515 |
| SVD-MLP-window | 0.564 | 0.549 | 0.573 | 0.603 | 45.071 | 167.140 |
| VAE-LGB-GKDE | 0.431 | 0.447 | 0.568 | 0.616 | 5.995 | 13.486 |
| VAE-LGB-STD | 0.431 | 0.447 | 0.568 | 0.616 | 11.015 | 24.836 |
| VAE-LGB-window | 0.431 | 0.447 | 0.568 | 0.616 | 17.510 | 79.773 |
| VAE-XGB-GKDE | 0.397 | 0.427 | 0.577 | 0.624 | 6.520 | 16.739 |
| VAE-XGB-STD | 0.397 | 0.427 | 0.577 | 0.624 | 11.978 | 30.368 |
| VAE-XGB-window | 0.397 | 0.427 | 0.577 | 0.624 | 18.883 | 97.485 |
| VAE-MLP-GKDE | 0.509 | 0.520 | 0.584 | 0.619 | 13.545 | 33.851 |
| VAE-MLP-STD | 0.509 | 0.520 | 0.584 | 0.619 | 24.770 | 61.455 |
| VAE-MLP-window | 0.509 | 0.520 | 0.584 | 0.619 | 38.593 | 196.246 |
| NAS-Bench-NLP11 | | | | | | |
| SVD-LGB | **0.906** | **0.882** | **0.862** | **0.820** | - | - |
| SVD-XGB | 0.849 | 0.865 | 0.786 | 0.735 | - | - |
| SVD-MLP | 0.120 | 0.108 | 0.292 | 0.275 | - | - |
| VAE-LGB | 0.789 | 0.795 | 0.802 | 0.747 | - | - |
| VAE-XGB | 0.826 | 0.838 | 0.797 | 0.739 | - | - |
| VAE-MLP | 0.150 | 0.160 | 0.315 | 0.300 | - | - |

Table 4: NAS-Bench-111 rank correlations computed on a separate test set with architectures trained for two different random seeds each. This allows the comparison with the rank correlation of an independent set of ground truth architectures. We find that the NAS-Bench-111 mean model is on par with the ground truth.

| Benchmark | Avg. $R^2$ | Final $R^2$ | Avg. KT | Final KT |
|---|---|---|---|---|
| NAS-Bench-111 | 0.557 | 0.541 | **0.660** | 0.860 |
| Ground truth (1 seed) | **0.593** | **0.920** | 0.619 | **0.873** |

Table 5: NAS-Bench-311 rank correlations computed on a separate test set with architectures trained for five different random seeds each. This allows the comparison with sets of learning curves averaged over multiple seeds. We find that the NAS-Bench-311 mean model performs better than a 4-seed mean.

| Benchmark | Avg. $R^2$ | Final $R^2$ | Avg. KT | Final KT |
|---|---|---|---|---|
| NAS-Bench-311 | **0.731** | 0.845 | **0.637** | **0.718** |
| Ground truth (1 seed) | 0.534 | 0.782 | 0.508 | 0.641 |
| Ground truth (mean of 2 seeds) | 0.651 | 0.835 | 0.555 | 0.683 |
| Ground truth (mean of 3 seeds) | 0.690 | 0.859 | 0.579 | 0.704 |
| Ground truth (mean of 4 seeds) | 0.710 | **0.870** | 0.592 | 0.712 |

explained in Section 3, the accuracies from the original NAS-Bench-101 benchmark were used as features to improve the performance of our surrogate, but not used as ground truth. In Table 4, we show that the KT values for NAS-Bench-111 are roughly equivalent to those achieved by a 1-seed tabular benchmark.

For NAS-Bench-311, training was straightforward. We used the original NAS-Bench-301 dataset, which already achieves good coverage [60], and we did not use any additional features. In Table 5, we show that the mean model in NAS-Bench-311 achieves higher rank correlation even than a set of learning curves averaged over four random seeds, by using a separate test set from the NAS-Bench-301 dataset which evaluates 500 architectures with 5 seeds each.

For NAS-Bench-NLP11, as described earlier, it is challenging to create an accurate surrogate benchmark because there are only $14\,322$ evaluated architectures for a search space of total size $10^{53}$. Therefore, we used two techniques to improve performance. First, we used a subset of the search space, restricting the architectures to a maximum of 12 nodes (reducing the size to $10^{22}$), and we added the validation accuracies from the first three epochs of training each architecture, as features. These two techniques were shown to substantially improve the performance of NAS-Bench-NLP11, as shown in Table 2. On an RTX 3070, training architectures from NAS-Bench-NLP takes about 90 seconds per epoch. Although adding in the first three epochs substantially improves the accuracy of our surrogate benchmark, it comes at the cost of query time. While NAS-Bench-111 and NAS-Bench-311 take under one second to query, a query to NAS-Bench-NLP11 now requires training an architecture for three epochs. Note that this is still a $15\times$ speedup over performing NAS directly without a surrogate benchmark.

We also create NAS-Bench-211 to further evaluate our surrogate creation technique (since NAS-Bench-201 already has complete learning curves). We train the surrogate on 90% of architectures from NAS-Bench-201 ($14\,062$ architectures) and test on the remaining 10%. SVD-LGB-window achieves the best performance. The rank correlation values are on par with a 1-seed tabular benchmark (see Table 6).

# D  Details from Section 5 (Experiments)

In this section, we give more details from Section 5, and we present more experiments.

Our work uses existing NAS Benchmarks. In Table 7, we report the licenses for each one.

Table 6: NAS-Bench-211 rank correlations computed on a test set with architectures trained for three different random seeds each. This allows the comparison with sets of learning curves averaged over multiple seeds. We find that the NAS-Bench-211 mean model performs on par with 1-seed ground truth.

| Benchmark | Avg. $R^2$ | Final $R^2$ | Avg. KT | Final KT |
|---|---|---|---|---|
| NAS-Bench-211 | 0.893 | 0.958 | **0.701** | 0.842 |
| Ground truth (1 seed) | 0.866 | 0.999 | 0.646 | 0.916 |
| Ground truth (mean of 2 seeds) | **0.900** | **0.999** | 0.679 | **0.926** |

Table 7: Licenses for the datasets that we use.

| Dataset | License | URL |
|---|---|---|
| NAS-Bench-101 | Apache 2.0 | https://github.com/google-research/nasbench |
| NAS-Bench-201 | MIT | https://github.com/D-X-Y/NAS-Bench-201 |
| NAS-Bench-301 | Apache 2.0 | https://github.com/automl/nasbench301 |
| NAS-Bench-NLP | None | https://github.com/fmsnew/nas-bench-nlp-release |

## D.1 LCE Results

Next, we give the LCE results for four search spaces, which is an extension of the results from Figure 3. That is, we test the improvement of three different single-fidelity algorithms when used with our LCE framework from Section 4, using WPM or SVR as the LCE techniques. We see that across all search spaces, for each single-fidelity algorithm, WPM and SVR both give improvements over the original algorithm, and SVR tends to give the larger improvement compared to WPM. In Figures 4 and 7, an earlier version of the NAS-Bench-NLP11 noise model was used. We also added slight clipping for NAS-Bench-111 and -311 to reduce the number of spike anomalies as described in Section 3.3.

## D.2 Ablation study.

We evaluate the effect of different fidelities on NAS-Bench-311. In Figure 8, we plot the validation regret of the SVR and WPM-based algorithms after $2 \times 10^6$ seconds, varying the initial fidelity (epoch) from which the learning curve is extracted, from 10 to 40. That is, the leftmost points run LCE by extrapolating from epoch 10 to epoch 100, and the rightmost points run LCE by extrapolating from epoch 40 to epoch 100. Note that there is a tradeoff between time saved (from only evaluating to 10 epochs vs 40) and accuracy of LCE (extrapolating from 10 epochs is more challenging than from 40 epochs). We see that overall, epoch 20 performs the best. Notably, BANANAS-SVR and REA-SVR (two of the best-performing algorithms across all search spaces) achieve top performance at epoch 20.

## D.3 NAS algorithm descriptions and details

We give a description and implementation details for each NAS algorithm from Section 5. Note that all algorithms were implemented in NASLib [56], keeping the implementation as close as possibe to the original implementation.

- **Random search.** Random search is a simple baseline which draws architectures at random and then returns the architecture with the lowest validation error. Note that multiple papers have shown that random search is competitive with other NAS algorithms [34, 57].

- **Local search.** Another baseline, local search has been shown to perform surprisingly well [68, 48], even on the DARTS search space [60]. It works by iteratively evaluating all architectures in the neighborhood of the current best architecture found so far. The neighborhood is defined as the set of architectures which differ by one operation or edge. We used the implementation from NASLib [56]. Notably, this is slightly different from the White et al. [68] implementation which may explain the worse performance on NAS-Bench-311.

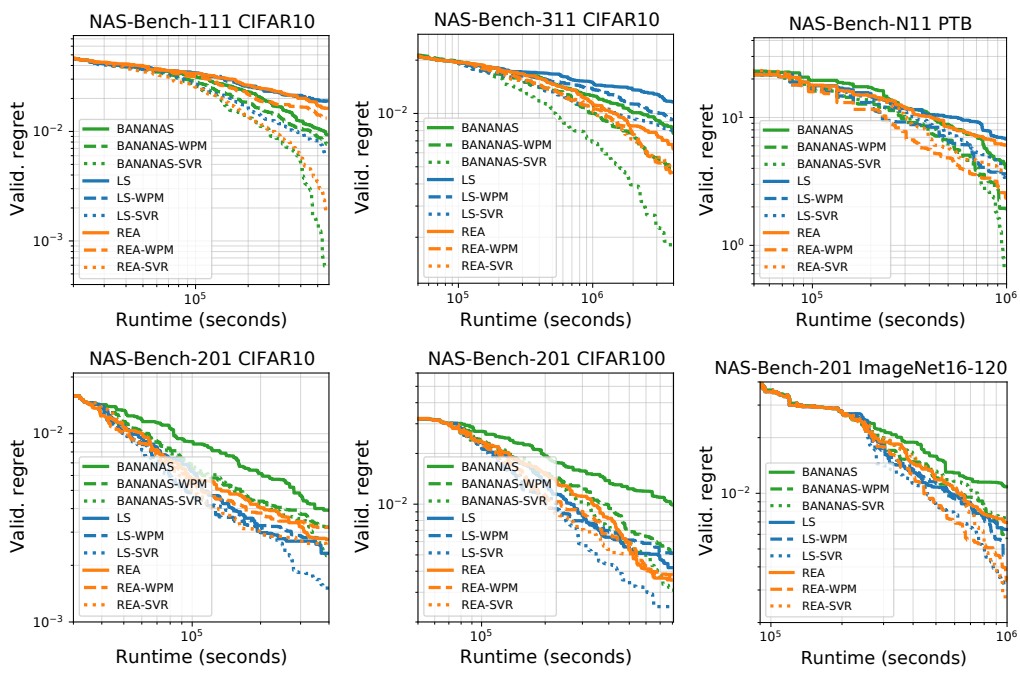

Figure 7: LCE Framework applied to single-fidelity algorithms on NAS-Bench-111, NAS-Bench-311, NAS-Bench-NLP11, and NAS-Bench-201.

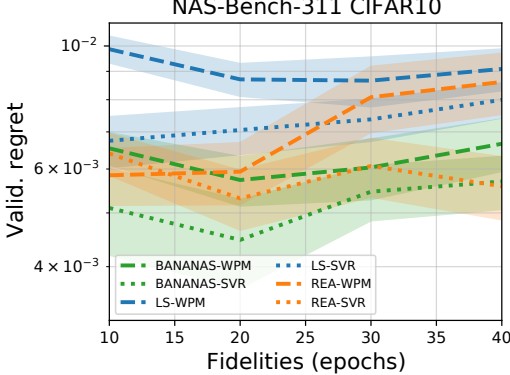

Figure 8: Different fidelities and their effect on NAS performance on NAS-Bench-311. The wall-clock time [s] is set to 2e6. The result are reported across 30 seeds.

- **BANANAS.** This algorithm [67] is based on Bayesian optimization using an ensemble of three MLPs as the model. We use the code directly from the original repository. We set the encoding to the adjacency matrix encoding instead of the path encoding. A predictor (trained on all architectures evaluated so far) chooses $k$ architectures which are then evaluated. In our experiments, the candidate pool is created by mutating the top four architectures ten times each (two times for each of the edit distance from one to five), and we set $k = 20$.

- **Regularized evolution.** This algorithm [53] is based on evolution. It consists of iteratively mutating the best architectures out of a sample of all architectures evaluated so far. A mutation is defined as randomly changing one operation or edge. We used the NAS-Bench-101 [76] implementation, changing the population size from 50 to 20.

- **Hyperband.** This algorithm [35] is based on random search with successive halving. It is based on successive halving, in which architectures are iteratively trained at a low fidelity, and then only the best-performing architectures are trained for longer in the next iteration, until the maximum number of epochs is reached. Hyperband performs multiple rounds of successive halving at different initial fidelities. We use the `hpbandster` implementation, adapted to NASLib [56].

- **Bayesian optimization Hyperband.** This algorithm [14] is based on combining Hyperband with Bayesian optimization. It starts the same way as Hyperband, but in the later rounds, for each fidelity a KDE model is trained using the trained architectures from previous rounds. Then the best architectures are chosen using Bayesian optimization with the model. We use the `hpbandster` implementation, adapted to NASLib [56].