# OpenReview forum: "NAS-Bench-x11 and the Power of Learning Curves"
_NeurIPS.cc/2021/Conference — NeurIPS 2021 Poster_

### Official Review · Reviewer_GXqH · 2021-07-11

**Rating:** 7
**Confidence:** 4

**Summary:**

Existing NAS benchmarks often only include the architecture performance at the final or some selected checkpoint epochs. However, per-epoch information is essential for developing and evaluating multi-fidelity NAS algorithms. This work proposed to fit a surrogate model to predict the learning curve given an architecture encoding. Utilizing the proposed predictor, the author extends three existing benchmarks NASBench-101, NASBench-301, and NASBench-NLP to include per-epoch accuracies. Further, to demonstrate the necessity of developing multi-fidelity algorithms for NAS, the author also describes a procedure to extend the single-fidelity algorithms to multi-fidelity ones by using learning curve extrapolation to filter out candidate proposals with partial training. Despite possible room for improvements on the performance of the learning curve predictor used, the proposed method would enable researchers to develop and rapidly test multi-fidelity NAS algorithms. I vote for acceptance.

**Limitations And Societal Impact:**

Yes

**Main Review:**

Strength:
1. The contribution of this paper is relevant to the NAS community and would accelerate research on multi-fidelity NAS algorithms.
2. The performance of the proposed learning curve predictor model is overall satisfactory.
3. The author adopted the proposed method to extend three existing benchmarks to include per-epoch information.

Weakness:
1. The author reported ranking correlation and KL divergence between the predicted learning curve and ground truth. The numbers seem reasonable, but the errors are still non-negligible. It is unclear how misleading these errors would be if someone tries to develop multi-fidelity algorithms by querying the proposed extended benchmark. How would the existing multi-fidelity NAS algorithms perform under the proposed benchmark, over ground-truth data? This might be doable for NASBench201 or NASBench111 (the version created without using information about the given 4 checkpoint epochs), as they provide ground-truth accuracies on at least some checkpoints?

Comments and Questions:
1. For NASBench-NLP, using the accuracy from the first few epochs as features significantly improves the learning curve predictor's performance. Can we also expect visible improvements on other spaces, if the info of the first few epochs is also provided?

2. You mentioned in para 355 that future surrogate works should save full learning curve info to avoid the need for any extra extensions. But surrogate benchmarks like NASBench-301 only train a small portion of the architectures fully, and the large majority part of the search space will be benched by the predictor. So it seems that such surrogate benchmark would still need to be extended with learning curve prediction anyway. Could you elaborate more on this?

**Time Spent Reviewing:**

4.5

---

> ### Author Response · Authors · 2021-08-10
> **We agree - we provide further validation for our surrogates**
>
> We thank the reviewer for their insightful and thorough review. We are pleased that the reviewer notes that our contributions will accelerate research on multi-fidelity NAS algorithms.
>
> We agree that the errors in our surrogate (Table 2) are hard to interpret and might be non-negligible. Therefore, we have conducted new experiments to further interpret and improve our surrogates. We give the full details in our comment above to all reviewers, and we give a summary here.
>
> In Table 2, rank correlations of 1.0 are not possible even for perfect models (or even tabular benchmarks), because of the natural noise in learning curves. Therefore, we computed a new statistic, tabular benchmark performance, which gives reference values to the Kendall Tau and $R^2$ statistics. We find that NAS-Bench-311 is very close to tabular benchmark performance, and NAS-Bench-111 is reasonably close to tabular benchmark performance. Since the submission deadline, we have also trained over 400 additional NAS-Bench-111 architectures to further improve the performance of NAS-Bench-111.
>
> Finally, it is a great suggestion to further evaluate our technique using NAS-Bench-201. We created a new surrogate, NAS-Bench-211, and we report its rank correlation compared to NAS-Bench-201. Please see the details in our comment to all reviewers above. We are currently running NAS algorithms on NAS-Bench-211 to compare to NAS-Bench-201.
>
> We now reply to the reviewer’s final "comments and questions."
>
> (1) *Will adding accuracies from the first few epochs help other surrogates?* This is a great question. We run this experiment on NAS-Bench-311, using the same test set from Table 2 of our paper.
>
> ||Avg. $R^2$|Final $R^2$|Avg. KT|Final KT|
> |---------------------------|--------|--------|--------|--------|
> |Original|0.798|0.859|0.721|0.780|
> |With first 3 epochs added|0.845|0.882|0.749|0.795|
>
> Based on these results and the results from NAS-Bench-NLP11, it looks like adding the first three epochs does substantially improve performance. This is a great option for NAS research that is a bit more computationally intensive but gives higher-quality results.
>
> (2) *Clarification about whether future benchmarks will need extensions.* Yes, the reviewer is correct that our learning curve prediction technique will be needed for all future NAS benchmarks that are too large to evaluate exhaustively, such as NAS-Bench-301. The only point we meant to make on line 355 is that for the unique case of NAS-Bench-101, they *did* evaluate exhaustively, but they neglected to save the entire learning curves (we emailed the author to double-check). We will update the paper to clarify this point.
>
> We thank the reviewer again for their review. If you have any more questions about our replies, or any further questions or concerns, please let us know.

---

> > ### Comment · Reviewer_GXqH · 2021-08-23
> > **Thank you for the reply**
> >
> > I appreciate the author's response and the extra experiments provided. There seems to be an interesting discussion going on with reviewer NYyv, the author's response on that thread also helped addressing some of my further questions.

---

### Official Review · Reviewer_NYyv · 2021-07-13

**Rating:** 5
**Confidence:** 5

**Summary:**

The authors describe a mechanism to predict learning curves which they use to generate surrogate benchmarks from existing NAS-Bench versions without learning curves. The authors explore different ways to create learning curve embeddings, different ways to predict these embeddings using the architecture representation as an input and different ways to model the training noise. Furthermore, the authors discuss a short study on using learning curve extrapolation in context of speeding-up NAS.

**Limitations And Societal Impact:**

Yes

**Main Review:**

The authors provide a good description of their method, alternative ways to solve each problem faced and support their decisions with empirical evidence.

My main concern is about the usefulness of this surrogate benchmark which has problems as partly discussed by the authors. It remains unclear what impact these problems would have.

The noise modelling in this paper leads to artifacts we would not see in real learning curves. First, it is very unlikely that the noise is architecture- and hyperparameter-independent. Second, given just the very few examples in Figure 1 we observe that the predicted learning curve in early epochs can already exceed to highest value of the entire learning curve (possibly due to very high noise levels).
In general, all surrogate benchmarks have the problem that they introduce some sort of bias which will not match what you would see in reality. Therefore, these datasets are more similar to synthetic datasets than real ones. They might be nice to prototype but I wouldn’t use them to evaluate NAS methods.

The experiment on combining learning curve extrapolation is a nice reminder for the NAS community to keep these approaches in mind when comparing different approaches. However, it provides no novelty since these methods have been used in the context of NAS already (see [1]).


**Time Spent Reviewing:**

4

---

> ### Author Response · Authors · 2021-08-10
> **We agree, and we have now added further evaluation**
>
> We thank the reviewer for their insightful review. The reviewer’s main concern is that our learning curve predictions are not accurate enough to be used for NAS experiments. We agree that this is a concern that should be taken seriously, and we will update the paper to reflect this. First, we respond to the specific issues that the reviewer noted, and then we will describe the steps we have taken to further evaluate and improve our surrogates.
>
> For one of our three noise models, we made an assumption that the noise is architecture-independent. The reviewer is correct that this assumption is unlikely to be quite true in real life. In our paper, we analyzed all assumptions that we made: in Appendix lines 722 - 739, and Table 3, we ran additional experiments and ablation studies that checked our assumptions. We also proposed three separate noise models with different assumptions, to give a higher chance that at least one model is reasonably accurate for any given search space.
>
> The reviewer correctly notes that in Figure 1, top left, the prediction in an early epoch exceeds the highest value. While this is an uncommon occurrence, this also happens sometimes in real-life learning curves. For example, we showed in Appendix Figure 6 that the noise in real-world learning curves is roughly Gaussian.
>
> In response to this review, we ran additional experiments to evaluate and improve our surrogates. In Table 2, rank correlations of 1.0 are not possible even for perfect models (or even tabular benchmarks), because of the natural noise in learning curves. We computed a new statistic, tabular benchmark performance, which gives reference values to the Kendall Tau and $R^2$ statistics. We find that NAS-Bench-311 is nearly tied with tabular benchmark performance, and NAS-Bench-111 is not too far off. We have also trained over 400 additional NAS-Bench-111 architectures, which increases its training data to improve performance further. Finally, we created a new surrogate, NAS-Bench-211, purely for evaluation purposes, to validate our learning curve prediction technique on yet another search space. Please see the full details of these experiments in our comment to all reviewers above.
>
> In conclusion, the reviewer is correct that we should understand and quantify the error in our surrogate benchmarks, especially before they are used by the NAS community. Our new experiments give further evidence that our surrogates perform on par with widely adopted tabular NAS benchmarks, and we will update the paper to be more explicit about the assumptions and remaining limitations.
>
> If you found our comments helpful, we respectfully ask that you consider increasing your score. Please let us know if you have any follow-up questions. We are happy to keep this discussion going.

---

> > ### Comment · Reviewer_NYyv · 2021-08-16
> > **Not all architectures are equal**
> >
> > The noise in real learning curves on NASBench101 can be Gaussian and yet there is no learning curve where the validation accuracy is after 40% of training higher than after 100%. Errors like this will always happen since the learned model cannot be perfect. The biggest problem is that even though the error is low on average, the error will be high for the maximum validation score (i.e. like an adversarial attack on the surrogate model) which is obviously the region of particular interest for NAS methods. This sort of behavior has been already described in completely different domains, e.g., Kumar et al.: Conservative Q-Learning for Offline Reinforcement Learning.
> >
> > Instead of weighing all architectures equally, errors for the learning curves with highest predicted validation accuracy (for each epoch) should be considered special since they get special treatment by NAS methods as well.
> >
> > Thank you for reporting additional experiments. I am not sure why any deviation from the tabular baseline in any direction, up or down, is acceptable. If they perform similar, there should be no deviation, right?

---

> > > ### Author Response · Authors · 2021-08-18
> > > **Thank you for your reply**
> > >
> > > We thank you very much for replying and keeping the discussion going! Specifically, your suggestion to weight the best architectures higher, significantly improved our surrogate. We respond to all of your comments below.
> > >
> > > *Not all architectures are equal.* We agree that not all architectures have the same noise distribution. We would like to clarify that as shown in Table 3 in our submission, the only search space that performed best with the GKDE model was DARTS, which is known to be an extremely homogeneous search space [1]. For NAS-Bench-111 and NAS-Bench-NLP11, the sliding window noise model works best (which does not make that assumption).
> > >
> > > [1] https://arxiv.org/abs/1912.12522, Section 5.1: “As we can observe from Figure 4, architectures sampled from [the DARTS search space] all perform similarly, with a mean of $97.03 \pm 0.23$.
> > >
> > > *No learning curve where the accuracy after 40% training is higher than after 100%.* Although this never happens for good architectures, NAS-Bench-101 contains all possible sets of operations in the search space, even strange ones consisting of only pooling operations or only identity operations, which can have uncommon learning curves. We iterated through NAS-Bench-101 and found 2,564 architectures out of 423k where the accuracy at epoch 36 is higher than the accuracy at epoch 108.
> > >
> > > *No deviation from a tabular benchmark is acceptable.* Yes, you are correct. First we give a quick note: a perfect tabular benchmark would train all architectures in the search space 100 times, so that researchers could run 100 fully independent trials of their NAS algorithm. NAS-Bench-101 and 201 train each architecture 3 times, so there is a chance that a few of the best architectures got 3 below-average seeds, which would lead to rank correlation errors. NAS-Bench-NLP trains each architecture just one time, so there may be even more rank correlation errors.
> > >
> > > A tabular benchmark that evaluates three seeds per architecture will have better rank correlation than a tabular benchmark that evaluates just one seed per architecture. We ran a new experiment to determine exactly how our surrogate compares to tabular benchmarks with 1 to 4 seeds. We find that our surrogate has the same rank correlation as a tabular benchmark with 3 seeds (the standard deviation is 0.02 on average).
> > >
> > > ||Avg. $R^2$|Final $R^2$|Avg. KT|Final KT|
> > > |-----------------------|--------|--------|--------|--------|
> > > |Tabular (1 seed)|$0.553$|$0.778$|$0.529$|$0.654$|
> > > |Tabular (2 seeds)|$0.672$|$0.845$|$0.581$|$0.709$|
> > > |Tabular (3 seeds)|$0.707$|$0.854$|$0.602$|$0.718$|
> > > |Tabular (4 seeds)|$0.727$|$0.870$|$0.617$|$0.732$|
> > > |NAS-Bench-311|$0.715$|$0.838$|$0.628$|$0.711$|
> > >
> > > *Give the architectures with the highest accuracy the highest weight.* Thank you, this is an excellent idea, and we can change the loss function to account for this. We tried weighting the architectures in the loss function two ways: $w_i = y_i - \min_j{y_j}$ or $w_i = 1/(\max_j y_j - y_i)$, and we found the first one worked best. We compared this new surrogate to the top half of architectures in our test set, which turned out to be much more challenging than the full test set, even for the tabular benchmark. Using your suggestion, the rank correlation increases significantly compared to our original surrogate and compared to the tabular benchmark. We attach the tables below.
> > >
> > > In brief, we addressed your main question by training a surrogate that has low error on the best architectures. If you have any new suggestions, we will be happy to implement and report.

---

> > > > ### Author Response · Authors · 2021-08-18
> > > > **Results for the top architectures**
> > > >
> > > > Interestingly, our original surrogate already ranked the top architectures better than the tabular benchmark. Using your suggestion, we see a significant improvement for the top architectures.
> > > >
> > > > ## Top architectures in the test set:
> > > >
> > > > ||Avg. $R^2$|Final $R^2$|Avg. KT|Final KT|
> > > > |-----------------------|--------|--------|--------|--------|
> > > > |Tabular (4 seeds)|$0.066$|$0.024$|$0.285$|$0.379$|
> > > > |NAS-Bench-311|$0.092$|$0.076$|$0.348$|$0.399$|
> > > > |Improved 311|$0.120$|$0.103$|$0.362$|$0.425$|
> > > >
> > > >
> > > >
> > > > ## All architectures in the test set:
> > > >
> > > > ||Avg. $R^2$|Final $R^2$|Avg. KT|Final KT|
> > > > |-----------------------|--------|--------|--------|--------|
> > > > |Tabular (4 seeds)|$0.727$|$0.870$|$0.617$|$0.732$|
> > > > |NAS-Bench-311|$0.715$|$0.838$|$0.628$|$0.711$|
> > > > |Improved 311|$0.699$|$0.825$|$0.633$|$0.717$|

---

> > > > ### Comment · Reviewer_NYyv · 2021-08-18
> > > > **Clarifications**
> > > >
> > > > What does it mean if your generated learning curves correlate with the groundtruth as much as the average learning curve over 3 seeds? Does this mean that your generated learning curves are less noisy? Does this make the NAS problem simpler?
> > > >
> > > > You report that roughly 0.6% of NASBench101 learning curves show better performance at 36 epochs than 108 epochs. Can you set this into perspective to how this looks like for your generated learning curves? What I expect is that for those NASBench101 curves the overall architecture performance is bad. Is this also the case for your generated learning curves? At least your example in the paper implies otherwise. I think there is a big difference between a learning curve that at first increases and then goes down again and a learning curve which has a very high spike in the middle but it has an improving trend overall.
> > > >
> > > > Your top architectures are still 50% of all learning curves? What I meant was about 10 architectures according to the maximum validation score over the entire learning curve. It is very likely that these are outliers.
> > > >
> > > > Giving a more weight to the better performing architectures was meant for evaluation purposes.

---

> > > > > ### Author Response · Authors · 2021-08-20
> > > > > **Responding to clarifications**
> > > > >
> > > > > Many thanks for your continued interest. We respond to your clarification comments below.
> > > > >
> > > > > *Does this mean that your generated learning curves are less noisy?*
> > > > > No, apologies for the confusion. As always, our goal is to output realistic (noisy) learning curves, but this experiment evaluates the predicted mean learning curves.
> > > > >
> > > > > *Overall performance is bad. Is this also the case for your generated learning curves?*
> > > > > You are right, we found that for 2551 of the 2,564 architectures, the final validation accuracy is below 0.65 (bottom 1.4% of NAS-Bench-101 accuracies), and our learning curves correctly predict this.
> > > > >
> > > > > *There is a big difference between a learning curve that at first increases and then goes down again and a learning curve which has a very high spike in the middle but it has an improving trend overall.*
> > > > > You are right. From all the architectures that we trained to get a full learning curve on NAS-Bench-101, 0.05% of real learning curves and 0.13% of predicted learning curves have at least one epoch less than 40 which has a higher validation accuracy than the final epoch. This is not a concern for our other surrogates, and we already reported in our [general comment](https://openreview.net/forum?id=V8PcLz1NoQ0&noteId=VR9hBdu8bGQ) that NAS-Bench-111 has a bit lower performance than tabular benchmarks, but adding more training data has already improved the $R^2$ to within 8% of tabular performance, and we are continuing to train more architectures to use as training data. But for now, we will add this to our section on limitations.
> > > > >
> > > > > *Giving more weight to the better performing architectures was meant for evaluation purposes.*
> > > > > Thank you, now we understand. Interestingly (as we briefly mentioned in our last comment), we found that our original surrogate already ranked the top 50% architectures better than the tabular benchmark. For your final question below, we will go back to using our original surrogate.
> > > > >
> > > > > *What I meant was about 10 architectures according to the maximum validation score over the entire learning curve.*
> > > > > You make a very important point, that the top 10 architectures are by far the most important architectures for NAS. Although, we respectfully point out that this is a concern even for NAS-Bench-101 and NAS-Bench-201 (since the learning curves are noisy and they only evaluate 3 times, there is a chance that a few of the best architectures get 3 below-average seeds).
> > > > >
> > > > > We report the results below. Our surrogate actually does beat tabular benchmark performance, but since the test set is size 10, the results are not statistically significant. We also compute the mean squared error because the rank correlations were too low to see conclusions.
> > > > >
> > > > > ||Avg. $R^2$|Final $R^2$|Avg. KT|Final KT|Avg. MSE|Final MSE|
> > > > > |-----------------------|--------|--------|--------|--------|--------|--------|
> > > > > |NAS-Bench-311|$-1.38$|$-12.7$|$0.061$|$0.114$|$0.64$|$0.04$|
> > > > > |Tabular (4 seeds)|$-1.96$|$-17.062$|$0.018$|$0.341$|$0.695$|$0.053$|
> > > > >
> > > > > In the future, we can train the best architectures from each search space 20 times each, to have more statistically significant results for this experiment.
> > > > >
> > > > > We are happy to respond to additional clarifications or follow-up concerns.

---

> > > > > > ### Comment · Reviewer_NYyv · 2021-08-24
> > > > > > **Clarification on top architectures**
> > > > > >
> > > > > > I am not sure we have a common understanding on the top 10 architecture topic. What is your definition of the best architecture?
> > > > > >
> > > > > > The point I am trying to make here is that when generating the entire validation/test learning curve, you might perform well if you only consider the final score. However, the best validation score might be observed after already 95% of the training such that one would like to use this checkpoint and use this point to report the test accuracy. Therefore, the best performing run is where the validation/test score is highest at any point of time. Why is this important? The learning curve is predicted and if you consider now the max value you might choose a learning curve with strongest predicted error. This can be a concern in general since methods such as Hyperband might then choose architectures with highest prediction error. What do I mean with highest prediction error? You report the MSE which might be low on average. However, there will individual cases where this error is very high.

---

> > > > > > > ### Author Response · Authors · 2021-08-25
> > > > > > > **Now we are on the same page**
> > > > > > >
> > > > > > > Thank you, now we understand your concern.
> > > > > > >
> > > > > > > *What is your definition of the best architecture?* In our last two comments, we had defined the top architectures as the *ground-truth* learning curves with the highest maximum validation accuracy over all epochs. Now we understand that your concern is with anomalies in the highest maximum accuracy for the *predicted* learning curves. We now came up with a few statistics to check whether or not our predicted learning curves have more anomalies than ground truth learning curves.
> > > > > > >
> > > > > > > First, in addition to the statistics that we already have, “Avg. KT” and “Final KT”, we add “Max. epoch KT”, to compare the rank correlation of architectures by the maximum validation accuracy that they achieved over the full learning curve. For NAS-Bench-311, max KT is 0.700. For the tabular benchmark, max KT is 0.696 (compared to final KT of 0.711 and 0.718, respectively).
> > > > > > >
> > > > > > > Next, we can compute the max-epoch MAE of the top ten predicted architectures (the ones with the highest validation accuracy anywhere on the curve), to the ground-truth results for these architectures. However, we note that we would need to train these architectures more times with different seeds to reach statistical significance. The average max MAE for the surrogate is 0.016, and for tabular it is 0.013.
> > > > > > >
> > > > > > > Finally, we can compare the percentage of architectures with “spike anomalies” of our predicted learning curves compared to the percentage of spike anomalies in the ground truth architectures. This is similar to the statistics we mentioned in our last comment, but now we will define it slightly differently: any architecture that has an accuracy at epoch <95 which is at least 1% higher than the accuracy at the final epoch. We find that on NAS-Bench-311, the rate of 0.148% is close to the ground truth rate of 0.111%. For NAS-Bench-111, the rate of 1.66% is higher than the ground truth rate of 0.64%. This matches our intuition: since NAS-Bench-311 has a large training set size (60k), it can match the noise distribution very well, while NAS-Bench-111 currently only has a smaller training set of full learning curves (1.5k) and does not predict the noise distribution as accurately. This is consistent with the conclusion in our last reply.
> > > > > > >
> > > > > > > Thank you for this conversation, since it has led to a more careful examination of our surrogates. We hope that we understood your concerns correctly now and that our experiments help to address them; if this is not the case, please don’t hesitate to clarify and ask for a different evaluation.

---

### Official Review · Reviewer_S4sg · 2021-07-16

**Rating:** 5
**Confidence:** 4

**Summary:**

Existing NAS benchmark datasets (e.g., NAS-Bench-101) do not provide full training information of architectures, which make it infeasible to evaluate the multi-fidelity algorithms (e.g., learning curve extrapolation). To address these issues, this paper create surrogate benchmark datasets, namely NAS-Bench-111, NAS-Bench-311 and NAS-Bench-NLP11, with singular value decomposition and noise modeling methods. Extensive experimental results with the full training information on these datasets show the superiority of the proposed method.

**Limitations And Societal Impact:**

The authors have addressed the limitations and potential negative societal impact of their work.

**Main Review:**

## Positive points:
1. The paper presents a method to extend the existing single-fidelity benchmarks to multi-fidelity ones with full training information.
2. The authors create NAS-Bench-111, NAS-Bench-311, and NAS-Bench-NLP11 benchmark datasets, which may be beneficial to develop new NAS algorithm.

## Negative points:
1. Does the proposed method apply to real world NAS application? In practice, given a dataset, we may have no available training information of any architectures. In this case, how to apply the proposed method? It would be limited if the proposed method is only able to extend the existing NAS benchmarks.
2. In line 220, why do the authors sample architectures from BANANAS, local search and regularized evolution?
3. In line 225, the authors mention that the method is able to improve the performance with the accuracies from the tabular benchmark (at epochs 4, 12, 36, 108) as additional features. More details of this should be provided.


**Time Spent Reviewing:**

5

---

> ### Author Response · Authors · 2021-08-10
> **Thank you, we will add these clarifications to the paper**
>
> We thank the reviewer for their helpful review. We reply to the three concerns below.
>
> 1. *Is our method only able to extend existing benchmarks?* As we explained on line 335, our method not only extends existing benchmarks, but can also be used to create new NAS benchmarks in the future. In this way, our technique is a new tool that the NAS community can use to create surrogate benchmarks for new real-world applications. We also proposed a new set of multi-fidelity NAS algorithms in Sections 4 and 5 (Algorithm 2) that are extremely useful in real-world NAS applications. For example, [auto-sklearn](https://github.com/automl/auto-sklearn) and [Auto-PyTorch](https://github.com/automl/Auto-PyTorch) are both popular AutoML tools that use Hyperband and BOHB, respectively, and we show in Figure 4 that our BANANAS-SVR and LS-SVR both outperform Hyperband and BOHB.
>
> 2. *Why do the authors sample from BANANAS, local search, and regularized evolution?* In line 220, we describe the method that we used to sample a set of architectures that has good coverage over the search space, with a focus on areas likely to be searched during NAS. To accomplish this, we trained *randomly sampled architectures* as well as *architectures from popular NAS algorithms* such as BANANAS, local search, and regularized evolution. This is the same technique used in prior work [1,2]. For example, [1] sampled architectures from BANANAS, regularized evolution, and other NAS algorithms (Section 3.1 in [this paper](https://arxiv.org/pdf/2008.09777.pdf)).
>
> 3. *Provide more details for our use of tabular benchmark accuracies as features.* Thanks for the feedback, we will make this clearer in the paper. When the original NAS-Bench-101 dataset was created, the authors trained all 423k architectures in the search space but only saved the accuracy of each architecture at epochs 4, 12, 36, and 108. Unfortunately, they did not save the other epochs (we confirmed via personal correspondence with the authors). Therefore, we trained a new set of 1100 architectures in order to have the accuracies for all epochs from 1 to 108. When we train the surrogate using the set of 1100 architectures, we can use the accuracies from the original NAS-Bench-101 at epochs 4, 12, 36, 108 as features, which substantially improves performance on the test set. We also note that for NAS-Bench-NLP11 and NAS-Bench-311, we see significant improvement when adding the accuracy from the first three epochs as features, which is practical in most cases. See our response to reviewer GXqH for more details.
>
> We will update the paper to make points (1), (2), and (3) very clear to future readers.
>
> Since we have replied to all of your questions, if you find our answers satisfactory, we respectfully ask that you please consider increasing your score. If you have any more questions or concerns, please let us know and we will be happy to answer.
>
> [1] https://arxiv.org/pdf/2008.09777.pdf
> [2] https://ojs.aaai.org/index.php/AAAI/article/view/9375

---

> ### Author Response · Authors · 2021-08-20
> **Additional questions**
>
> Please let us know if you have any follow-up questions or concerns. We would be happy to answer them.

---

### Official Review · Reviewer_xYsu · 2021-07-16

**Rating:** 6
**Confidence:** 3

**Summary:**

This paper proposes another benchmark dataset for NAS, NAS-Bench-x11. Compared to the original benchmarks NAS-Bench-101, NAS-Bench-301 and NAS-Bench-NLP, NAS-Bench-x11 contains the full training
information for each candidate architecture especially the learning curves so that it allows multi-fidelity NAS algorithms

NAS-Bench-x11 is a surrogate benchmark, which trains a surrogate model to predict the learning curve of the vast amount of candidate architectures. Prediction of the learning curve can be accomplished by singular value decomposition and noise modeling.


**Limitations And Societal Impact:**

Yes

**Main Review:**

Strength:

The paper is well motivated and clearly written. The paper is clear about limitations too.

I think this paper will be valuable to the community.  The created NAS-bench-x11 contains the full training information of different architectures, which may make it more convenient to analyse NAS algorithms and promote further NAS algorithm design. The idea of predicting the learning curve is very interesting to me.

The authors conducted extensive experiments to demonstrate the effectiveness of the proposed method.

Weakness:

This paper is a completion of the previously created benchmarks. Providing learning curves is also considered before in NAS-Bench-201, which weakens the novelty of this paper. So I recommend a weak acceptence.


**Time Spent Reviewing:**

7

---

> ### Author Response · Authors · 2021-08-10
> **Thank you for your review**
>
> We thank the reviewer for mentioning that the paper is well-motivated, clearly written, and valuable to the community. We agree that predicting learning curves is an interesting question.
>
> We agree that NAS-Bench-201 having learning curves slightly weakens the novelty, but we respectfully point out that NAS-Bench-201 only evaluated 15625 (6466 unique) architectures and saved the learning curves. However, *very large* benchmark datasets of size $>10^{18}$ are much more valuable to the community, since they are more challenging and more realistic. Before our paper, it was not possible to have learning curves for these large benchmarks. We present a novel technique to predict learning curves on large benchmark datasets.
>
> Please let us know if you have any follow-up questions, clarifications, or concerns about our work.

---

> > ### Comment · Reviewer_xYsu · 2021-08-24
> > **The author provides satisfactory responses**
> >
> > I appreciate the author's endeavor during rebuttal and the experiments provided. After reading the authors' response and discussing with other reviewers, I decided to keep my score.

---

> ### Author Response · Authors · 2021-08-20
> **Follow-up questions**
>
> If you have any follow-up questions or concerns, please let us know. We would be happy to answer them.

---

### Author Response · Authors · 2021-08-10
**Additional evaluation of our surrogates**

Dear reviewers and AC, thank you for the valuable feedback. We have added the clarifications and suggestions mentioned by the reviewers, and we think this has improved the clarity and overall contributions of our paper. We give a list of the main changes below, which focus on further evaluating and improving our surrogates. Please let us know if you have any questions about the new additions.

(1) First, we added a new reference value for the metrics in Table 2 called "tabular benchmark performance". The idea is that in Table 2, we computed the rank correlation of our surrogate predictions to a set of ground-truth (noisy) learning curves. Because of the random noise, even a perfect surrogate model (or even a tabular benchmark) would not achieve Kendall Tau and $R^2$ values of 1.0. In order to provide a reference value, we compute the rank correlation that a tabular benchmark would achieve. Note that common tabular benchmarks (NAS-Bench-101, NAS-Bench-201) evaluate every architecture on three random seeds, so for each architecture, it predicts on a fourth (independent) seed by averaging the accuracies of its three random seeds. This is a very strong reference value: while our surrogates have no access to any learning curves from the test set, the tabular benchmark statistics average three learning curves per architecture on the test set.

For NAS-Bench-311, we used the "de-multi-seed" set from NAS-Bench-311 as our test set because all architectures were trained 5 times each. Since we used a new test set, the numbers are a bit different from Table 2. Our experiment shows that NAS-Bench-311 nearly matches tabular benchmark performance. In fact, for Avg. KT, it *outperforms* tabular benchmark performance (and there is precedent for surrogate benchmarks outperforming tabular benchmarks -- see Section 2 of [NAS-Bench-301](https://arxiv.org/pdf/2008.09777.pdf)).

||Avg. $R^2$|Final $R^2$|Avg. KT|Final KT|
|-----------------------|--------|--------|--------|--------|
|Tabular|0.727|0.870|0.617|0.732|
|NAS-Bench-311|0.715|0.838|0.628|0.711|

(2) For NAS-Bench-111, we have been continuing to train architectures since the NeurIPS submission deadline. We have now increased the training set for NAS-Bench-111 by about 30%. We use a separate test set of random architectures trained five times. Especially with the additional data, NAS-Bench-111 approaches tabular benchmark performance.

||Avg. $R^2$|Final $R^2$|Avg. KT|Final KT|
|-----------------------|--------|--------|--------|--------|
|Tabular|0.664|0.919|0.643|0.869|
|NAS-Bench-111|0.581|0.639|0.636|0.769|
|w/ additional data|0.610|0.900|0.657|0.850|

(3) From the suggestion of reviewer GXqH, we created NAS-Bench-211. The purpose of this is purely to evaluate our technique, because NAS-Bench-201 already has the full learning curve information. We created NAS-Bench-211 by splitting up the NAS-Bench-201 architectures into training and testing data in a 90-10 split. We train a surrogate, and then we compute the rank correlation statistics below. This is a more challenging setting because there are only 6466 unique architectures in all of NAS-Bench-201, but our model still approaches tabular benchmark performance.

||Avg. $R^2$|Final $R^2$|Avg. KT|Final KT|
|-----------------------|--------|--------|--------|--------|
|Tabular|0.824|0.998|0.624|0.911|
|NAS-Bench-211|0.823|0.943|0.651|0.785|

Please let us know if you have any additional concerns or clarifications. We are looking forward to having a discussion in the coming days.

---

> ### Author Response · Authors · 2021-09-01
> **Thank you**
>
> We would like to thank all reviewers once again for the initial reviews and lively discussions during the rebuttal period, which has helped to further improve and refine our paper.

---

### Decision · Program_Chairs · 2021-09-27

**Decision:**

Accept (Poster)

**Comment:**

This paper adds learning curve information to popular NAS benchmarks via a surrogate model. This is valuable to the community since it opens up current NAS benchmarks to multi-fidelity search approaches. Furthermore it demonstrates a compeling 'template' that can be used for creating new NAS benchmarks on new search spaces where it is not feasible to create tabular benchmarks due to the sheer size of the search space.

During the review process a number of improvements have come about with the reviewers, especially when engaging with NYyv with respect to top architecture predictions. It is strongly recommended that the authors include this new information in the paper. In this vein, the paper should clearly mark the limitations of this benchmark and how NAS researchers should consume this benchmark and interpret the outcomes of their search algorithms.

This benchmark is going to be widely used by the community. So the quality of the software release and ease-of-usage is quite critical. Please pay a lot of attention to making sure that the package can be installed and run easily via standard Python package distribution channels like pip/conda and the associated data/surrogate models can be installed in one-click and there are plenty of examples to showcase how to use the benchmark.